# HPV16 and HPV18 Genome Structure, Expression, and Post-Transcriptional Regulation

**DOI:** 10.3390/ijms23094943

**Published:** 2022-04-29

**Authors:** Lulu Yu, Vladimir Majerciak, Zhi-Ming Zheng

**Affiliations:** Tumor Virus RNA Biology Section, HIV Dynamics and Replication Program, Center for Cancer Research, National Cancer Institute, National Institutes of Health, Frederick, MD 21702, USA; lulu.yu@nih.gov (L.Y.); majerciv@mail.nih.gov (V.M.)

**Keywords:** papillomaviruses, genome structure, RNA splicing, RNA polyadenylation, epigenetic modification

## Abstract

Human papillomaviruses (HPV) are a group of small non-enveloped DNA viruses whose infection causes benign tumors or cancers. HPV16 and HPV18, the two most common high-risk HPVs, are responsible for ~70% of all HPV-related cervical cancers and head and neck cancers. The expression of the HPV genome is highly dependent on cell differentiation and is strictly regulated at the transcriptional and post-transcriptional levels. Both HPV early and late transcripts differentially expressed in the infected cells are intron-containing bicistronic or polycistronic RNAs bearing more than one open reading frame (ORF), because of usage of alternative viral promoters and two alternative viral RNA polyadenylation signals. Papillomaviruses proficiently engage alternative RNA splicing to express individual ORFs from the bicistronic or polycistronic RNA transcripts. In this review, we discuss the genome structures and the updated transcription maps of HPV16 and HPV18, and the latest research advances in understanding RNA cis-elements, intron branch point sequences, and RNA-binding proteins in the regulation of viral RNA processing. Moreover, we briefly discuss the epigenetic modifications, including DNA methylation and possible APOBEC-mediated genome editing in HPV infections and carcinogenesis.

## 1. Introduction

Human papillomaviruses (HPV) are a group of small, non-enveloped DNA viruses, 52–55 nm in diameter. To date, the group consists of more than 450 genotypes (types) [1,2,3], in which DNA sequences in the most conserved L1 open reading frame (ORF) differ by more than 10%, even from the closest known HPV type. About 40 HPV types can infect the genital tract and ~80% of the sexually active population will contract a genital HPV infection at some point in their lives. Based on the association with cancer, genital HPV types can be divided into low-risk types, causing mainly genital warts, and high-risk types, which may cause invasive cancer. Almost 100% of cervical cancers, 25–100% of other types of genital cancers, including anus, penis, vagina and vulva, as well as ~40–70% of head and neck cancers, can be attributed to high-risk HPV infection [4,5]. Among the ~15 types of high-risk HPVs that include HPV16, 18, 21, 33, 35, 39, 45, 51, 52, 56, 58, 59, 68, 73, and 82, HPV16 and HPV18 are the two most common types, accounting for ~70% of all HPV-related cervical cancers [6]. In contrast to low-risk HPVs, high-risk HPVs express two potent oncoproteins, E6 and E7, that mediate, respectively, degradation of cellular p53 and pRb, two tumor suppressor proteins essential for cell cycle control [7,8] and genome stability [9,10,11], leading to HPV-induced carcinogenesis.

HPV infection is cell-type-specific. HPV infects keratinocytes on the basal lamina through micro-wounding induced by scratching or sexual intercourse. Initial attachment of HPV virions to the host keratinocytes takes place by the interaction of viral capsid proteins with host cell receptors [12]. The virus in the infected keratinocytes replicates in a differentiation-dependent manner [13,14], with its early genes, including viral oncogenes E6 and E7, being expressed in undifferentiated or intermediately differentiated keratinocytes, while the expression of the late genes occurs only in the keratinocytes undergoing high or terminal differentiation. This highly complex, cell-differentiation-dependent expression of viral genes results from using alternative viral promoters, RNA splicing, and RNA polyadenylation to ensure a proper time and expression level of viral proteins at each viral replication stage. This review will focus on HPV16 and HPV18, the two major oncogenic HPVs, for their genome structures and expression. We will also update the current knowledge of their transcriptomes and the roles of cellular factors in regulating their RNA processing.

## 2. Papillomavirus Genome Structure and Expression

Papillomaviruses contain a double-stranded, circular DNA genome of ~8 kb in size. All papillomavirus genomes can be separated by two polyadenylation (pA) signals, viral early (pA_E_) and viral late (pA_L_), into three distinct regions: an upstream regulatory region (URR), an early (E), and a late (L) gene regions (Figure 1a). The URR region does not encode any protein and is also known as a long control region (LCR). The size of the URR varies among papillomaviruses (Table 1), being 853 bp in HPV16 and 825 bp in HPV18. The HPV URR contains the binding sites for viral E1 and E2 proteins for initiating HPV replication, and several cellular transcription factors, including NFI, Oct-1 (OCT), AP-1, TEF-1 (TF1), and SP1, for transcription initiation [15]. The URR also contains an origin of replication (Ori) which is crucial for papillomavirus replication and is highly homologous to the mammalian autonomous replicating conserved sequences (ACS) [16] (Figure 1b). Each papillomavirus genome contains a single Ori responsible for viral DNA replication.

The HPV early region encodes seven viral non-structural proteins: E1, E2, E1^E4, E5, E6, E7, and E8^E2 (Figure 1a). E1 is a viral Ori-binding DNA helicase with E2 being an E1 accessory protein. Thus, both E1 and E2 are essential for papillomavirus DNA replication. E2 also acts as a viral transcriptional activator or repressor, depending on where it binds, and participates in virus genome segregation during mitosis [19,20]. E5, E6, and E7 are viral oncogenes inducing cell immortalization and transformation. Although the HPV E5 function in cell immortalization or transformation is not well defined, viral E6 and E7 target several oncogenic and tumor suppressor proteins [8,21] and miRNAs [22] to abrogate the cell cycle checkpoints. Specifically, E6 interacts with cellular tumor suppressor p53 and E7 with pRb [23]. E8^E2 protein inhibits viral gene expression and viral genome replication [24]. Interestingly, E1^E4 encoded from the viral early region is a viral non-structural late protein expressed in the late stage of infection to facilitate the virus egress by disrupting the host cytokeratin network [25,26,27]. The late region of the HPV genome encodes two viral capsid proteins: L1 and L2 (Figure 1a). Expression of L1 and L2 at the late stage of infection is restricted to highly, or terminally, differentiated epithelial cells and is triggered by viral vegetative DNA replication [28]. As all ORFs are located on the same strand of the double-stranded viral genome, papillomavirus transcription is unidirectional, occurring only from one strand of the viral genome. There is no antisense transcription during virus infection even though some background (noise) antisense transcripts may be detectable [29,30]. All viral promoters, named by their transcription start site, are in the URR and early regions of the virus genome. The number of the identified promoters varies among papillomaviruses (Table 1). Papillomaviruses do not encode their own RNA polymerase and utilize host RNA polymerase II [31] for their transcription. Thus, all viral promoter activities are subject to regulation by host and viral transcription factors and chromatin modifications. The viral transcripts derived from each promoter are then polyadenylated, either at an early pA_E_ or a late pA_L_ site, using host polyadenylation machinery [32,33]. HPV early promoters and pA_E_ are active during the early stage of the viral life cycle, whereas viral late promoters and pA_L_ become active during the late stage of the viral life cycle. E1^E4 transcripts are initiated from the late promoter P_L_ but are normally polyadenylated at the pA_E_ site to encode the E1^E4 protein.

Due to a relatively small genome and the compacted genomic organization, papillomaviruses transcribe their RNAs as bicistronic or polycistronic mRNAs with a 5′ m^7^G cap to encode multiple viral proteins. However, based on the current dogma of 5′ cap-dependent eukaryotic translation starting from the RNA 5′ to 3′ direction and in the absence of an internal ribosome entry site (IRES) in papillomaviral RNA transcripts, only the first ORF located at the transcript’s 5′ end is efficiently translatable (Figure 2a), whereas the downstream ORFs not translatable become a part of the transcript’s 3′ untranslated region (UTR). In some cases, a leaky scanning 40S ribosome might skip the first ORF AUG with weak Kozak content [34,35] and thus allow recognition of the downstream ORF AUG (Figure 2b). All the first three ORFs (papillomaviral E6, E7, and E1 ORFs) of papillomavirus polycistronic transcripts have good Kozak content; therefore, a leaky scanning model might not be a plausible mechanism for viral protein translation from papillomavirus polycistronic RNAs.

Evolutionarily, papillomaviruses have developed the use of alternative promoters to transcribe viral RNAs with an alternative first ORF. An excellent example of this strategy is using alternative promoters to express E6 RNA from one promoter and E7 RNA from another promoter by animal papillomaviruses [29] and low-risk HPVs [36,37]. However, high-risk HPVs have developed several alternative RNA processing mechanisms to express their proteins from a single polycistronic RNA transcribed from a single promoter (Figure 2c). First, high-risk HPVs utilize alternative RNA splicing for the RNA transcripts derived from a single early promoter to express E6 and E7. Second, utilization of alternative RNA polyadenylation sites, in combination with alternative RNA splicing, allows the expression of L1 and L2 from a late promoter located in the early region. Third, use of alternative translational frames generated by alternative RNA splicing facilitates the expression of viral E1^E4 or E8^E2 from the RNA transcripts with the coding regions partially overlapping with E2 or E1 ORF, respectively (Figure 2d). As a result, the HPV transcriptome consists of a large number of alternatively processed RNAs. This highly complex regulation of viral RNA processing ensures the desired level of individual viral proteins at a proper time for a complete productive cycle of virus infection. A dysregulation of these RNA processing steps is a hallmark of HPV-induced carcinogenesis. To date, we have comprehensively mapped the transcriptome of only a handful of human and animal papillomaviruses.

## 3. Genome Expression and Transcription Maps of HPV16 and HPV18

As more than 90% of HPV16 and HPV18 genomes encode ORFs, alternative RNA splicing of intron-containing bicistronic or polycistronic RNAs bearing more than one ORF to generate mRNA transcripts is essential for the favorable expression of a downstream ORF. Similar to all other papillomaviruses, all introns in HPV16 and HPV18 bicistronic or polycistronic transcripts have been characterized as major U2-type GU-AG, not minor U12-type AU-AC, introns [38,39,40].

### 3.1. HPV16 Genome Expression and Transcription Map

As a leading causative agent for cervical cancer, HPV16 genome transcription and viral oncogene E6 and E7 expression were the major focus of HPV studies in the past ~40 years. The HPV16 transcription map was initially generated by multiple laboratories based on viral RNAs extracted from cervical cancer tissues, HPV DNA-transfected cells, and HPV-infected tumor cell lines containing both extrachromosomal and integrated HPV16 genomes [41,42,43]. The full HPV16 transcription map illustrating sixteen transcripts, currently at the Papillomavirus Episteme website (PaVE) (https://pave.niaid.nih.gov/, accessed on 1 March 2022), is an old, outdated version from the Human Papillomaviruses Compendium-1996 composed by Carl Baker and Charles Calef (https://pave.niaid.nih.gov/lanl-archives/HPVcompintro4.html, accessed on 1 March 2022). This old version was then updated in 2006 [44] to include an additional four viral transcripts. In this review, new features discovered in the past 16 years since 2006 are collected in an updated transcription map which includes a total of twenty-three viral transcripts and two newly identified promoters P14 and the putative P1135 (Figure 3).

In general, HPV16 genome expression is driven mainly by two major promoters. The early promoter P97 upstream of the E6 ORF contains a TATA box (a eukaryotic core promoter motif) 35-nt upstream of its transcription start site and is used for transcription of nearly all viral early gene transcripts, while the late promoter P670 within the E7 ORF is a TATA-less promoter responsible for the expression of HPV late genes. The HPV16 E2 protein regulates the promoter P97 activity through binding to the cis-elements in the URR region [45,46,47]. HPV16 P97-derived early pre-mRNA transcripts undergo alternative RNA splicing to generate various isoforms of mRNAs (b-o) to produce viral early proteins. Eight major RNA splicing sites, including two 5′ splice donor sites (5′ ss or SD) at nt 226 (SD226) and nt 880 (SD880), and six 3′ splice acceptor sites (3′ ss or SA) at nt 409 (SA409), nt 526 (SA526), nt 742 (SA742), nt 2582 (SA2582), nt 2709 (SA2709), and nt 3358 (SA3358) are used for alternative RNA splicing to generate viral early mRNA isoforms (Figure 3, RNA species a–o). The viral late RNA transcripts are transcribed from a late promoter P670, highly activated by viral vegetative DNA replication in highly differentiated keratinocytes. These late viral transcripts are also spliced to produce five isoforms of mRNAs (Figure 3, species p–t). The majority of the late pre-mRNAs are spliced from SD880 to SA3358 to produce E1^E4 transcripts (Figure 3, RNA species q), but only a small fraction of these late transcripts is double-spliced from SD880 to SA3358 and then from SD3632 to SA5639 to produce L1 transcripts (Figure 3, RNA species s). Occasionally, the SD880 may be spliced to SA2709 (Figure 3, RNA species p) or directly to SA5639 (Figure 3, RNA species t). The SD880 in the P97 transcripts might also splice back to the SA409 crossover the E7 ORF, producing a circular RNA circE7 with extremely low abundance (<0.4 copies per cell) as a side product of conventional RNA splicing [48]. This scarce amount of circE7 is not thought to have any claimed biological function [48]. Transcripts from a P14 promoter [49] and a putative P1135 promoter [50] are also detectable. The P14 promoter transcript was proposed to encode E1 after splicing from SD226 to SA409 [49] (Figure 3, RNA species a), whereas the P1135 promoter transcript was proposed to encode E8^E2 by splicing from SD1302 to SA3358 [50] (Figure 3, RNA species u). Other unusual RNA splicing patterns described in HPV16 quasivirion-infected keratinocytes [51], and rare viral promoters described in differentiated W12 subclone cells [52], remain to be further verified for possible future inclusion.

The HPV16 genome contains two canonical pA signals (AAUAAA) to guide host RNA polyadenylation machinery to cleave the viral RNA transcripts for the addition of a poly A tail and termination of viral RNA transcription. The pA_E_ at nt 4215 is mainly used for polyadenylation of viral early RNA transcripts, and the pA_L_ at nt 7321 is responsible for the polyadenylation of viral late RNA transcripts. However, the late RNA transcript E1^E4 uses the pA_E_ for its polyadenylation. The cleavage sites for RNA polyadenylation of HPV16 early transcripts have been mapped mainly to nt 4232 [53] and of late transcripts to nt 7345 [52].

HPV16 uses all three coding frames to translate viral proteins. Viral E6, E2, E8^E2, and L1 are expressed from the frame 2 and viral E7, E1, E5, and L2 from the frame 1. Interestingly, the first 5 aa of E1^E4 are translated from the frame 1 and are identical to the first 5 aa of E1 protein, but the rest of the E1^E4 protein is translated from the late RNA exon 2 in the frame 3 generated by viral RNA splicing (Figure 3). The transcript a derived from the P14 promoter has an in-frame AUG start codon at position nt 83, upstream of the annotated E6 start codon at nt 104, but the AUG at nt 83 is weaker than the AUG at nt 104 in the Kozak sequence context (ANNaugN or GNNaugG) [54,55], lacking a purine in position -3 of the AUG and a G in position +4 of the AUG for ribosome binding. Thus, this upstream AUG is unlikely to be useful for translation initiation. As shown in Figure 3, the annotated viral ORFs encoding viral E6, E1, and L2 overlap an intron, and thus, their ORF integrity is disrupted by nuclear RNA splicing in the infected host cells. How these three viral transcripts escape host RNA splicing machinery remains a mystery. In particular, the viral L2 expression requires the late transcript r both to escape the recognition of the early pA_E_ signal and the intron 2 splicing.

### 3.2. HPV18 Genome Expression and Transcription Map

The most recent updated HPV18 transcription map [56] is based on transcripts generated from productive HPV18 infection in primary human keratinocytes [38] and from U2OS cells transfected with an HPV18 plasmid [57] (Figure 4). In this transcription map, HPV18 shares many features with HPV16 in terms of transcription, alternative RNA splicing, and polyadenylation to express viral early and late genes. HPV18 has two major early promoters P55 and P102 (previously called P105) in the URR region, with a TATA box 27-bp and 25-bp upstream of transcriptional start sites, respectively. As the P55 TATA-box is located in the core Ori region and surrounded by three E2-binding sites, the P55 promoter activity can be inhibited by viral DNA replication, thus leading to a switch to the promoter P102 of which the TATA box lies outside of the Ori core [58]. In contrast to viral early promoters, the HPV18 late promoter lacking a consensus TATA box exhibits heterogeneous transcription start sites ranging from nt 590 to nt 907 in the virus genome. The late promoter P811 within the E7 ORF has been identified as a major HPV18 late promoter [38]. Its activity depends on vegetative viral DNA replication in highly differentiated keratinocytes, and it can be regulated by host factors binding to viral cis-acting DNA elements [28,59,60]. Two minor promoters might exist in the E1 (P1193/1202) [57] and E2 (P3036/3385) [57,61] ORF regions, but remain to be further verified. The P3036/3385 promoter can be activated in the differentiated keratinocytes [60].

HPV18 early transcripts originated from P55 or P102 are polyadenylated at the pA_E_ at nt 4235 with a predominant cleavage site at nt 4270 [38]. Similar to HPV16 early transcripts derived from the promoter P97, HPV18 early transcripts derived from the promoter P55/P102 undergo alternative RNA splicing using eight major splice sites, including two 5′ donor sites (SD233 and SD929) and six 3′ acceptor sites (SA416, SA791, SA2779, SA3434, SA3465, and SA3506), to generate an array of single- or double-spliced mRNAs for expression of various viral early proteins (Figure 4, RNA species a–h, j–k, and o–p). Among these splice acceptor sites, the SA416 and SA3434 are preferably used. The removal of intron one within the E6 ORF through alternative RNA splicing leads to disruption of E6 ORF integrity and promotes expression of E6*I, E6*II, E6*III, E7, and E6^E7 proteins. Similarly, alternative RNA splicing of the intron 2 in these transcripts disrupts the E1 ORF integrity but promotes expression of E2 and E5 in addition to E6 and E7. In this case, only the RNA transcript (Figure 4, RNA species a) that retains the intron 2 by escaping RNA splicing could express viral E1 protein. The promoter P1202 is a minor promoter [38,57] and its RNA transcripts bear only a single intron, which appears to be responsible for the expression of viral E8^E2, E5, and E2, through alternative RNA splicing from SD1357 to SA3434, SA3465, or SA2779 (Figure 4, RNA species i, n, and r).

HPV18 late transcripts initiated from the late promoter P811 can be polyadenylated using an early pA_E_ signal at nt 4235 for production of viral late protein E1^E4 (Figure 4, RNA species s), or a late pA_L_ signal at nt 7278 for expression of viral capsid proteins L1 (Figure 4, RNA species u, w–y) and L2 (Figure 4, RNA species t). The cleavage sites for polyadenylation of HPV18 L1 and L2 RNAs have been mapped to either nt 7299 or nt 7307 [38]. Thus, HPV18 late RNA transcripts could be both single intron- and double intron-containing RNA transcripts depending on which pA site is selected for RNA polyadenylation and transcription termination. In total, HPV18 late transcripts utilize eight RNA splicing sites, including three splice donor sites SD929, SD3696, and SD3786, and five splice acceptor sites SA3434, SA3465, SA3506, SA5613, and SA5776, to produce nine species of the viral late RNA transcripts (Figure 3, RNA species m, q, s–y), with the RNA species s, u, and y being three predominant late RNA isoforms. Intron 1 in HPV18 late transcripts spans partially the E1 and E2 ORFs and its efficient removal by constitutive RNA splicing during the late stage of virus infection leaves no detectable intron 1-containing late transcripts. However, intron 2 covers the early pA_E_ site and almost the entire L2 ORF. Intron 2 retention (Figure 3, RNA species t) by escaping both pA_E_ recognition and RNA splicing, which mechanism remains to be understood, is required for the expression of L2 protein.

## 4. Characteristics of Papillomavirus Introns and RNA Cis-Elements in the Regulation of Viral Alternative RNA Splicing

Eukaryotic RNA splicing [62] is carried out in the nucleus by a spliceosomal complex composed of five uridine-rich small nuclear RNAs (snRNA U1, U2, U4, U5, and U6) and ~200 associated proteins [63,64]. All introns are characterized by a 5′ ss interacting with U1, a branch point sequence (BPS) interacting with U2, and a 3′ ss interacting with U2AF [62]. A polypyrimidine tract (PPT), 15–40 nts in size, lies between the BPS and 3′ ss and binds PPT-binding proteins, such as PTB. RNA splicing begins with intron recognition in a stepwise interaction, leading to the removal of the recognized intron by two transesterification reactions and joining two exons by the U4/U6. U5 tri-snRNP complex [65,66]. The removed intron in a circular lariat structure is then debranched for degradation [67,68,69]. When a splice site or BPS is suboptimal, the splicing efficiency of this intron is subject to regulation by many cellular splicing factors, including serine/arginine (SR)-rich protein family members and heterogeneous nuclear ribonucleoproteins (hnRNPs) [70].

Almost all introns in papillomavirus transcripts are suboptimal [38,71,72,73]. RNA 5′ ss in both HPV16 and HPV18 transcripts miss base-pairing of a few nucleotides to the six core nucleotides of the U1 5′ end in the first step of RNA splicing [38,72]. Their 3′ ss are also weak, with multiple purines in the PPT, and are accompanied by a non-consensus BPS [74,75], and are therefore prone to an alternative selection of different 3′ ss. To understand how this alternative 3′ ss selection could occur in the course of papillomavirus infection, early studies using bovine papillomavirus type 1 (BPV-1) late pre-mRNAs, which contain a late leader 5′ ss and two alternative 3′ ss, a proximal 3′ ss at nt 3225 and a distal 3′ ss at nt 3605, showed that the alternative selection of the two suboptimal 3′ ss is regulated both by exonic splicing enhancers (ESE) and an exonic splicing suppressor or silencer (ESS) [76,77]. Selection of the proximal 3′ ss is controlled by an ESE-ESS bipartite regulator, SE1-ESS1 and SE2, upstream of the distal 3′ ss for expression of L2 [71,73,76], but usage of the distal 3′ ss, also with a suboptimal BPS, is under control by another ESE-ESS bipartite regulator SE4-ESS2, mainly confined to highly differentiated keratinocytes [77,78]. Selection of the distal 3′ ss splicing triggers RNA splicing of a downstream intron which covers the early pA site and partial L2 ORF, leading to the production of viral L1 mRNA [78]. Subsequent studies demonstrated that these RNA cis-elements (SE1, SE2, SE4, ESS1, and ESS2) function by interacting with host SR proteins and hnRNPs, leading to a viral early-to-late switch of the splice sites [79]. These interactions include AG-rich SE1 and SE2 binding SRSF1 (ASF/SF2, SRp30a), SRSF2 (SC35, SRp30b), SRSF4 (SRp75), and SRSF6 (SRp55) [80], the AC-rich SE4 interacting with SRSF3 (SRp20) [79], and pyrimidine-rich ESS1 interacting with PTB, U2AF, and SR proteins [81]. These groundbreaking discoveries set up a foundation for further explorations of how human papillomavirus transcription and post-transcriptional RNA processing in the infected keratinocytes are regulated for the translation of viral proteins in a cell differentiation manner.

Subsequently, a subset of RNA cis-elements has been discovered in the corresponding and other regions of HPV16 and HPV18 genomes that contribute to the regulation of HPV16 and HPV18 alternative RNA splicing [56,79,82,83]. Similar to BPV-1, these viral RNA cis-elements in HPV16 and HPV18 function by binding various host splicing factors, of which SR proteins bind to ESEs and hnRNPs to ESS in general [70]. In the following paragraphs, we will specifically discuss their roles in selecting individual alternative 3′ ss.

### 4.1. Intron 1 in the E6 ORF and Its Splicing Regulation of HPV16 and HPV18 Early RNA Transcripts

In contrast to animal and human low-risk HPVs, which express E6 and E7 from two separate promoters and the E6 ORF region which has no intron, high-risk HPVs transcribe E6 and E7 from a single major promoter as a polycistronic E6E7 pre-mRNA (Figure 3 and Figure 4). This E6E7 pre-mRNA contains three exons and two introns, with intron 1 being in the E6 ORF and intron 2 overlapping partially with the E1 and E2 ORFs. This feature of each intron leads to the alternative selection of different 3′ ss and thereby produces distinct RNA splicing isoforms. Only the intron 1-retained E6E7 RNA has an intact E6 ORF and thus translates a full-length E6 protein. The limited intercistronic space between the E6 stop codon and the E7 start codon only 2 nt apart in HPV16 and 8 nt in HPV18, prevents the translating ribosomes from translation re-initiation of the downstream E7 ORF after finalizing E6 translation (Figure 5a). In contrast, removal of intron 1 leads to a frameshift to create a premature stop codon, thus increasing the intercistronic space between newly formed E6*I ORF and E7 ORF to 144 nt for HPV16 and 130 nt for HPV18, of which intercistronic space is sufficient for the translating ribosomes to re-initiate translation of the downstream E7 ORF (Figure 5b). HPV16 E6*I ORF encodes a labile polypeptide with 43 amino acid residues [74] of which function remains to be investigated.

Intron 1 splicing and production of E6*I RNA responsible for efficient expression of viral E7 protein was repeatedly observed experimentally [72,84]. Ample evidence supporting these initial observations was provided in 2006 by using small interfering RNAs (siRNAs) specifically targeting the intron 1 in the E6 ORF in cervical cancer cell lines and further confirmed by E6E7 minigene assays [85]. There is no evidence to date whether E7 is translated through ribosome leaky scanning in cervical cancer cells or high-risk HPV immortalized cells, although an in vitro study was reported [86].

In addition to its suboptimal nature of splice sites, HPV16 E6E7 intron 1 has a BPS heptamer AACAAAC with single adenosine (underlined) at nt 385 to serve as a branch site [74] during the first step of RNA splicing reactions (Appendix A). HPV18 E6E7 intron 1 exhibits two overlapped, identical BPS heptamer AACUAAC, where one heptamer has a branch site adenosine (underlined) at nt 384 and the other at nt 388 [75], for E6*I splicing (Appendix A). In exploring the regulation of E6*I RNA splicing, recent studies demonstrated that an RNA cis-element ESS functions as a negative RNA splicing regulator. This ESS was first discovered in the HPV18 E7 ORF in 2016 [56] and later in 2020 in the corresponding region of HPV16 E7 ORF [87]. It is positioned between nt 612–639 in the HPV18 genome and between nt 594–604 in the HPV16 genome. It bears a UUAGA core motif that binds hnRNP A1 to inhibit the intron 1 (233^416) splicing of HPV18 E6E7 pre-mRNAs [56] or the intron 1 (226^409) splicing of HPV16 E6E7 pre-mRNAs [87], thus allowing expression of the viral E6 protein. Knockdown of hnRNP A1 expression in the cells increases E6*I RNA splicing and thereby the production of E7 proteins [56,87] (Appendix A). Additionally, EGF signaling via Erk1/2 activation appears to inhibit HPV16 226^409 splicing of E6E7 pre-mRNAs in the HPV16 E6E7-immortalized keratinocytes and HPV16-positive SiHa but not CaSki cells [88], suggesting a possible cell-type regulation of E6*I RNA splicing. However, the downstream protein target(s) of Erk1/2 in the immortalized or SiHa cells remain to be carefully investigated.

### 4.2. A Common Intron in the E1 ORF of Both Early and Late Transcripts and Its Splicing Regulation in HPV16 and HPV18 Gene Expression

Intron 2 in the HPV16 early transcripts has one 5′ ss at nt 880 and three alternative 3′ ss, respectively, at nt 3358, 2709, and 2582 in the order of their frequency of usage in RNA splicing, with 880^3358 splicing being the most common splicing event [89] (Figure 3). In HPV18, intron 2 of the early transcripts has one 5′ ss at nt 929 and four alternative 3′ ss, respectively at nt 3434, 3465, 3506, and 2779, with 929^3434 splicing being the most common splicing event [38,56] (Figure 4). Because of its suboptimal features with an unknown BPS, this intron splicing is subject to extensive regulation by alternative RNA splicing. It is evident that intron 2 removal in the early transcripts by RNA splicing between 880^3358 of HPV16 or 929^3434 of HPV18 interrupts the integrity of both E1 and E2 ORFs and blocks the production of E1 and E2 proteins but creates an E1^E4 ORF in the spliced early transcripts. However, this newly created E1^E4 ORF is presumably not translatable from the early transcripts due to limited intercistronic space (only 7 nts) between the E7 stop codon and the E1^E4 start codon. However, it might serve for encoding the E5 protein in addition to the spliced E5 RNA isoform f in HPV16 (Figure 3) and h in HPV18 (Figure 4). To date, we know little about how the viral early transcripts escape intron 2 splicing to express the E1 protein in HPV-infected cells. However, the RNAs with 880^2709 splicing in HPV16 and 929^2779 splicing in HPV18 have been predicted to be the most likely to encode the viral E2 protein.

In the viral late transcripts, intron 2 of the viral early transcript is referred as intron 1 and is removed by RNA splicing from most late mRNAs transcribed from a late promoter P670 in HPV16 and P811 in HPV18. The viral late transcripts with 880^3358 splicing in HPV16 and 929^3434 splicing in HPV18 are predominant and necessary to encode viral late proteins of E1^E4 and L2, although alternative splicing of 880^5639 in HPV16 and 929^5613 in HPV18 can be detected for L1 production (Figure 3 and Figure 4). In addition, it has been well-documented that intron 1 splicing in the viral late transcripts commonly triggers late RNA intron 2 splicing between 3632^5639 in HPV16 and 3696^5613 in HPV18. The intron 2 splicing deletes the unwanted pA_E_ signal and disrupts L2 ORF integrity to produce the L1 protein. Notably, the generation of an L2 mRNA needs to bypass the pA_E_ recognition and retain intron 2 to reach the pA_L_ for L2 RNA polyadenylation.

As described above, alternative selection of 3225 3′ ss and 3605 3′ ss in BPV-1 is mediated through five RNA cis-elements (SE1, ESS1, SE2, SE4, and ESS2) by interaction with various RNA splicing factors [71,73,76,77,79,80,81,90]. Similar splicing regulation of an equivalent 3358 3′ ss in HPV16 and 3434 3′ ss in HPV18 has been observed and extensively studied. A 65-nt AC-rich splicing enhancer element at nt 3462–3527, promoting the selection of 3358 3′ ss usage for 880^3358 splicing in HPV16 [83], was first described in 2005 and further characterized as a BPV-1 SE4-equivalent 29-nt long ESE at nt 3488–3516 region [79], which binds SRp20 (SRSF3), SRp30s including SRSF1 (ASF/SF2 or SRp30a), SRSF2 (SC35 or SRp30b), 9G8 (SRSF7), and SRp30c (SRSF9), SRp55 (SRSF6), SRp75 (SRSF4), YB-1, and hnRNP L [79] (Appendix A). Later, an additional 9-nt AG-rich ESE (ACCGAAGAA) at nt 3446–3454 also enhances 880^3358 splicing by competitive binding of SRSF1, hnRNP G, and Tra2B [91,92] (Appendix A).

Although the AC-rich ESE binding SRSF3 and AG-rich ESE interacting with SRSF1, hnRNP G, and Tra2B are important for the usage of the 3358 3′ ss splicing of the HPV16 early transcripts to express viral E6 and E7 [79,91,93], the binding of SRSF3 to this AC-rich splicing enhancer also inhibits intron 2 3632^5639 splicing of the viral late transcripts to promote RNA polyadenylation at the pA_E_ for the viral E1^E4 expression, thus decreasing HPV16 L1 expression [79]. Inhibition of intron 2 3632^5639 splicing of the HPV16 late RNAs by SRSF3 appears mechanistically different from suppression by an hnRNP D and hnRNP A2/B1-binding inhibitory cis-element (AUAGUA) at nt 3600–3605 upstream of the 3632 5′ ss [94] (Appendix A). Like the 3358 3′ ss in HPV16, the 3434 3′ ss in HPV18 is the major 3′ ss for both early and late pre-mRNAs [38]. SRSF3 promotes HPV18 929^3434 splicing and E1^E4 production by interaction with the ESE from nt 3520–3550 in the HPV18 genome [56] (Appendix A).

### 4.3. Intron 2 in the L2 ORF and Its Splicing Regulation of HPV16 and HPV18 Late RNA Transcripts

Papillomavirus late RNA transcripts derived from the viral late promoter have an intron 2 spanning over a large part of the L2 ORF. This intron, featuring suboptimal splice sites and an unmapped BPS, needs to be retained to express L2 in the late stage of papillomavirus infection. Otherwise, removal of the intron 2 by RNA splicing of the late transcripts would not only delete the pA_E_ signal from the late mRNAs but also disrupt the integrity of the L2 ORF. Intron 2 3632^5639 splicing in the HPV16 genome or 3696^5613 splicing in the HPV18 genome is exclusively used for L1 production even though a few minor splice sites were occasionally detected in the virus-infected cells. The HPV16 late pre-mRNAs transcribed from its late promoter P670 are usually spliced once from nt 880^3358 for E1^E4 and L2 expression or twice from nt 880^3358, and then from nt 3632^5639 before reaching the late pA_L_ at nt 7321 for HPV16 L1 expression (Figure 3). Similarly, the HPV18 late pre-mRNAs transcribed from its late promoter P811 are spliced once from nt 929^3434 for E1^E4 and L2 expression, or twice from nt 929^3434, and then from nt 3696^5613 before reaching the late pA_L_ at nt 7278 for HPV18 L1 expression (Figure 4). In both HPV16- and HPV18-infected cells, only a small proportion of the viral late transcripts undergo double RNA splicing. Most of the late transcripts are spliced once and polyadenylated at the pA_E_ for E1^E4 production, with only a few transcripts capable of reading through the intron 2 5′ ss and the pA_E_ site for L2 expression (Figure 3 and Figure 4). 

Regulation of intron 2 splicing in the HPV16 late transcripts is a complicated process and involves both in RNA cis-elements adjacent to each splice site and host RNA-binding proteins (RBP). As shown in Figure 6a, the expression of HPV16 L1 mRNAs or intron 2 3632^5639 splicing could be inhibited by hnRNP L, hnRNP D, and hnRNP A2/B1 through binding to AUAGUA RNA motifs (nt 3600–3605) located 32 nt upstream of the 3632 5′ ss (Figure 6a) [94,95] (Appendix A). Interestingly, this suppression of the 3632 5′ ss usage could be relieved by transient overexpression of hnRNP C1, or its family member RALYL, which also bind to the same AUAGUA silencer element [96], or by an Akt-kinase-inhibitor-mediated reduction in hnRNP L-binding to this silencer element [95] (Appendix A). PTB binds to a 191-nt AU-rich intronic element at a nt 3653–3850 region downstream of the 3632 5′ ss, and overexpression of PTB was found to promote the 3632 5′ ss splicing and L1 expression [97] (Appendix A). Compared to the nt 3632 5′ ss, regulation of the nt 5639 3′ ss is much simpler. Downstream of this 3′ ss reside two ESS elements, a 55-nt ESS1 at nt 5639–5693 [95,98] and a 49-nt ESS2 at nt 5816–5864 in the HPV16 genome [82] (Appendix A). Both ESS1 and ESS2 bind hnRNP L, hnRNP A1, PTB, U2AF, and SRSF1 [95] to inhibit the selection of nt 5639 3′ ss for L1 expression (Appendix A). As all these RBPs are abundant in cells, in vivo binding of the HPV16 late RNAs to specific RBPs and knockout of individual RBPs in the testing cells for the L1 expression and productive HPV infection remain to be further investigated.

## 5. Papillomavirus RNA Polyadenylation and Its Regulation

The 3′ end processing of pre-mRNAs transcribed by RNA polymerase II is essential for RNA stability, export, and translation. RNA polyadenylation initiated by cleavage of the nascent transcript is coupled with the addition of non-templated 150–200 adenylate residues to an RNA transcript to form a poly-A tail. Four core cis-elements in the nascent transcripts, including a UGAU cis-element for a downstream pA signal (PAS) definition, a highly conserved AAUAAA hexamer representing PAS, a cleavage site generally positioned 10–30 nt downstream of PAS, and a G/U- or U-rich element (downstream element, DSE) that is ~10–30 nt further downstream of the cleavage site, are recognized by cellular polyadenylation machinery to initiate the process of RNA cleavage and polyadenylation [99,100]. Numerous cellular factors are involved in this process, including the cleavage and polyadenylation specificity factor (CPSF) that binds to the AAUAAA hexamer, the cleavage stimulation factor (CstF) that binds to the G/U- or U- rich sequence, the cleavage factors I and II (CFI and CFII) that bind to the cleavage site, poly-A polymerase (PAP) for poly A addition, and the poly A-binding protein (PABP) that associates with the nascent poly A tail to facilitate RNA export and translation [101,102,103,104].

### 5.1. Regulated RNA Polyadenylation at the Viral pA_E_ Site

All papillomaviruses have two polyadenylation signals, pA_E_ and pA_L_, for their gene expression. As shown in Figure 3 and Figure 4, HPV16 has a pA_E_ at nt 4215 for cleavage and polyadenylation at nt 4232 for all viral early and late E1^E4 RNA transcripts and a pA_L_ at nt 7321 for cleavage and polyadenylation at nt 7345 for viral late L1 and L2 RNAs. In parallel, HPV18 utilizes a pA_E_ at nt 4235 for cleavage and polyadenylation at the nt 4270 cleavage site and a pA_L_ at nt 7278 for cleavage and polyadenylation at the nt 7299/7307 site. Recognition of individual viral PAS sites is intimately coupled to viral promoter activation and alternative RNA splicing and thus linked to keratinocyte differentiation, with the pA_E_ being recognized in both undifferentiated and differentiated keratinocytes while the pA_L_ is recognized only in highly differentiated cells. Although not yet fully understood, additional RNA cis-elements upstream and downstream of the pA_E_ are involved in the regulation of viral pA_E_ recognition in HPV16 and HPV31 [105,106] (Figure 6a). More specifically, HPV16 applies two RNA cis-elements, a 57-nt U-rich element at nt 4155–4212 upstream of the pA_E_ [53] and a 96-nt triple-G-rich element at nt 4383–4479 downstream of the pA_E_ [107], to promote the pA_E_ activity (Appendix A). This enhancement of the pA_E_ usage is carried out by the 57-nt U-rich element interacting with hFip1, CstF64, hnRNP C1/C2, and PTB [53] (Figure 6a), and the 96-nt triple G-rich element-binding hnRNP H [107] (Appendix A). Interestingly, both the N-terminal and hinge regions of HPV16 E2 contribute to inhibition of pA_E_ recognition by the CPSF complex and were shown to cause a read-through at the pA_E_ into the late region to induce the expression of L1 and L2 mRNAs in an in vitro reporter study [108].

HPV18 contains six UGUA motifs in the 80-nt sequences upstream of its pA_E_ at nt 4235. These UGUA motifs are responsible for CFIm binding and polyadenylation efficiency [109,110]. It would be interesting to know whether the U-rich sequences upstream, and the G-triplets downstream of the HPV18 pA_E_, play similar roles as described for the polyadenylation of the HPV16 early transcripts.

### 5.2. Regulated RNA Polyadenylation at the Viral pA_L_ Site

Viral late pA_L_ usage is also regulated by RNA cis-element(s) and RBPs. The cis-elements controlling the activity of the viral late pA_L_ were initially discovered in BPV-1 late transcripts. These studies revealed a 53-nt BPV-1 inhibitory cis-element at the late 3′ UTR, which contains a consensus cryptic 5′ ss and inhibits late RNA polyadenylation at the late pA_L_. Through binding U1 snRNAs, this cryptic 5′ ss-containing cis-element inhibits late RNA polyadenylation by a direct interaction of U1-70K with poly(A) polymerase [111,112]. The 3′ UTR of HPV16 late RNAs also contains a 79-nt negative regulatory element (NRE) spanning from the last few nucleotides of the L1 ORF into the viral late 3′ UTR [113] (Appendix A). The HPV16 NRE proximal half contains four weak 5′ ss binding U1 snRNP-like complex. Its distal half is GU-rich and binds cellular factors U2AF65, HuR, SRSF1, CstF64 and CUG-binding protein 1 (CUGBP1) [114,115,116,117,118,119] (Figure 6a) (Appendix A). As the NRE is highly conserved in all other papillomaviral L1 RNAs, the functional consequence of U1 snRNP binding to these late 3′ UTRs is most likely to protect viral L1 RNAs from premature cleavage and polyadenylation as described [120,121].

Analyses of the 80-nt sequences 5′ to the HPV18 pA_L_ at nt 7278 showed five UGUA motifs in this region [38]. Similar to the HPV16 late 3′ UTR, the HPV18 late 3′ UTR contains multiple U1-binding sites (cryptic 5′ ss), including one from nt 7217 to 7225 upstream of the pA_L_ and four overlapped U1-binding sites (nt 7292–7302, nt 7300–7310, nt 7309–7319, nt 7321–7331) downstream of the late cleavage sites [38]. Collectively, they may suppress RNA polyadenylation of HPV18 late transcripts via their interaction with the factors identified for the HPV16 late NRE [114,115,116,117,118,119].

## 6. RBPs and Their Dysregulation by HPV Infections

The human genome encodes ~1500 RBPs [122,123,124]. As stated above, many RBPs involved in RNA splicing, polyadenylation, stability, and translation have been characterized in the regulation of papillomaviral gene expression at the post-transcriptional levels, thus contributing to the control of viral early and late stages of infection. However, these regulations are not unidirectional; the expression levels of RBPs are reciprocally affected by HPV infection.

### 6.1. Host RBPs in HPV Infections

One group of the common RBPs consists of abundant multifunctional hnRNPs [125,126]. Multiple hnRNPs, including hnRNP A1/A2, hnRNP C1/C2, hnRNP H, and hnRNP I (also referred to as polypyrimidine tract-binding protein 1, PTBP1), are significantly upregulated in HPV-induced high-grade lesions and cervical cancer tissues [127,128,129,130,131]. Their contributions to HPV carcinogenesis were assumed, as knockdown of hnRNP A1/A2 [132] and hnRNP I (PTBP1) [133] expression was found to prevent HeLa cell proliferation. In addition, the expression of hnRNP E1 (also referred to as poly C-binding protein 1, PCBP1) is inversely related to the progression from low-grade lesions to cervical cancer [134]. As well as hnRNPs, HPV-positive cervical cancer tissues show the altered expression of many other RBP genes, including increased expression of CDKN2A, ELAVL2, GRB7, HSPB1, KHSRP, and RNASEH2A and decreased expression of NOVA1 [130]. The increased expression of CDKN2A and RNASEH2A, and the decreased expression of NOVA1, are easily detectable from HPV16- and HPV18-infected keratinocytes [130]. RNASEH2A interacts with the proliferation cell nuclear antigen (PCNA) to remove the RNA primers and promote Okazaki fragment maturation during DNA replication [135]. NOVA1 (neuro-oncological ventral antigen 1) is a neuron-specific RNA-binding splicing regulator, which binds to the YCAY elements of the target pre-mRNAs to selectively enhance or suppress exon exclusion [136].

Host RNA splicing factor SR proteins, including SRSF1(ASF/SF2), SRSF2 (SC35), SRSF3 (SRp20), and SRSF10 (SRp38/SRrp40), are also upregulated in HPV-positive cervical precancer lesions and cancer tissues [127,137,138,139] (Figure 6b). These upregulations could be partially attributed to viral E2′s DNA binding and transactivation activities [140] and E6 and E7 protein activities [130]. Among them, SRSF1 (ASF/SF2) and SRSF3 (SRp20) are two proto-oncoprotein splicing factors [138,141,142], but their expressions are mutually dependent by reciprocal regulation of RNA splicing [143]. In cervical basal cells, SRSF1 and SRSF3 are co-expressed along with other splicing factors including SRSF2, hnRNP K, hnRNP L, hnRNP A1, and YB-1 [143]. Consistent with the observation that SRSF3 promotes the E1^E4 880^3358 splicing in HPV16 and 929^3434 splicing in HPV18, but suppresses viral L1 3632^5639 splicing in HPV16 and 3696^5613 in HPV18 [56,79], the expression level of SRSF3 in keratinocytes in raft cultures supporting L1 expression and virion production is much lower than in the keratinocytes in monolayer cultures which do not normally support L1 expression and production of infectious virions [79]. Importantly, the reduced expression of SRSF3 also leads to increase keratinocyte differentiation and L1 expression in human foreskin keratinocytes [56] and in papillomavirus-infected tissues [79]. Therefore, differentially expressed SRSF3 controls the HPV early-to-late switch (Figure 6b, left panel).

### 6.2. Viral Proteins Function as RNA-Binding Proteins in the Regulation of RNA Splicing and Polyadenylation

To investigate which viral protein(s) in HPV16 and HPV18 infections causes the increased expression of host RBPs, HPV16 E2, which binds DNA and regulates viral and host gene transcription, has been demonstrated to enhance the expression of SRSF3 (SRp20), SRSF1 (ASF/SF2), SRSF2 (SC35) and SRSF4 (SRp75) and SR proteins kinase 1 (SRPK1) [137,144,145,146]. HPV16- and HPV18-positive cancers display increased SRSF10 (SRp38/SRrp40) expression which could be upregulated by E6E7 via E2F1 transcriptional activation [139]. As a result, the increased RNA splicing factors in HPV-infected cells not only regulate viral RNA splicing [140], but also RNA splicing of host gene transcripts [147].

In addition to their DNA-binding activities, purified HPV16 E2 and E6 are two viral RNA-binding proteins which suppress in vitro RNA splicing via the E2 N-terminal half and hinge regions and the E6 N-terminal and central portions, respectively [148]. Both E2 and E6 preferentially bind to the intron region of pre-mRNAs via the E2 C-terminal DNA-binding domain and E6 nuclear localization signal-3 (NLS-3) in the E6 C-terminal domain [148]. HPV16 E2 interacts with SRp30s, SRrp40 and SRp75, whereas HPV16 E6 interacts with SRp30s and SRp75. Both E2 and E6 bind only weakly with SRp55 [148]. Although HPV5 E2 interacts with ASF/SF2, SC35, U1-70K, and U5-100K, through its hinge region containing multiple arginine-serine (RS) repeats and regulates RNA splicing of a reporter RNA [149], it has no effect on RNA splicing in vitro [148]. E2 proteins from HPV1, HPV8, HPV11, HPV16, and HPV18 interact with SRPK1 and E2 proteins from HPV1 and HPV8 also strongly bind SRPK2 [150]. SRPK1 and SRPK2 are two SR protein kinases important for SR protein phosphorylation to function [151,152]. Interestingly, HPV1 E1^E4 also interacts with SRPK1 and inhibits phosphorylation of SR proteins and HPV1 E2 [153]. PI3K/Akt3 and protein kinase C (PKC) also phosphorylate SR proteins and regulate papillomavirus RNA splicing and late gene expression [154,155].

HPV16 E2 inhibits polyadenylation in vitro by preventing the assembly of the CPSF complex at the pA_E_ site, which is reversible by high levels of HPV16 E1 protein or by the polyadenylation factor CPSF30. Both the N-terminal and hinge regions of HPV16 E2 contribute to this inhibition and cause a read-through at the pA_E_ into the late region, leading to induction of the expression of L1 and L2 mRNAs with increased L1 RNA 3632^5639 splicing [108].

## 7. DNA Methylation and APOBEC-Mediated Genome Editing in HPV Infections

### 7.1. DNA Methylation in HPV Infections

DNA methylation, catalyzed by the enzymatic activity of DNA methyltransferases (DNMTs), occurs predominantly on cytosines preceding guanine nucleotides (CpG dinucleotides) and maintains CpG methylation patterns in the cells following DNA replication [156]. Methylation of DNA is involved in the regulation of gene expression, and this regulation is context-dependent. In general, methylation in regulatory sequences of genes, such as promoters and enhancers, often leads to transcriptional silencing of genes, whereas transcription-active genes often have hypomethylated or unmethylated regulatory sequences. However, methylation within transcriptional silencers in gene bodies can enhance transcription and RNA splicing [157]. Infection with oncogenic HPVs enhances the expression and enzymatic activity of DNMT1, a major mammalian DNA methylation enzyme, leading to aberrant methylation patterns of both viral and host genomes after DNA replication. As p53 is capable of binding and repressing the DNMT1 promoter in cooperation with SP1 [158], HPV16 E6-mediated p53 degradation results in upregulation of DNMT1 expression [159]. E2F transcription factors activate the DNMT1 promoter [160]. HPV16 E7-mediated free E2Fs from associated pRb enhance DNMT1 expression [161]. HPV16 E7 also promotes DNMT1 activity by binding DNMT1 directly in vitro and in vivo via the E7 zinc-finger CR3 motif [162]. Thus, CpG methylation patterns of host genes in HR-HPV infections are actively investigated as possible specific biomarkers for HPV-related pre-cancer progression [163,164,165]. In HR-HPV-infected cervical tissues, EPB41L3, EDNRB, LMX1, DPYS, MAL, PAX1, and CADM1 genes were found to be highly methylated in CIN2 and CIN3 (CIN2/3) [166,167]. To date, EPB41L3 [164,168,169] and PAX1 [167,170] have been further used in a few large population studies.

All papillomavirus genomes contain hundreds of CpG dinucleotides, as shown in Table 1, and the correlation between DNA methylation and HPV genome expression has been extensively studied [171]. Methylation of the E2-binding sites (E2BSs) in the URR region reduces E2 binding and deregulates E6 and E7 expression [172,173]. CpG methylation in HPV L1 and L2 genes is associated with HPV pathogenesis and disease stage [174,175,176,177]. Methylation of HPV16, HPV18, HPV31, and HPV45 L1 and L2 genes and the cellular DAPK gene appears lowest in asymptomatic infection and increases successively along with cervical lesion progression to cancer [178]. Currently, methylation patterns of the high-risk HPV late regions and one of the selected host genes EPB41L3, PAX1, or DAPK, which is sensitive to high-risk HPV infection, are under active evaluation as possible biomarkers for diagnosis and prognosis of HPV-related pre-cancer progression [163,164,169,170].

### 7.2. Genome Editing and APOBEC3 Expression in HPV Infections

As well as DNA methylation on both host and viral genomes, host RNA-editing by deamination enzymes also plays an important role in genome modifications by introducing point mutations into RNA or single-stranded DNA (ssDNA). Adenosine-to-inosine (A-to-I) editing on double-stranded RNA (dsRNA) is catalyzed by dsRNA-specific adenosine deaminase ADAR1 or ADAR2 [179,180] of which ADAR1 exhibits an increased expression in cervical lesions and cancer tissues [181,182], whereas cytidine-to-uridine (C-to-U) editing on a single-stranded RNA (ssRNA) and single-stranded DNA involves 11 members of the mammalian cytidine deaminase family, the apolipoprotein B messenger RNA-editing enzyme catalytic polypeptide (APOBEC) family [183,184]. Among seven members of APOBEC3 (A3) proteins (A3A, A3B, A3C, A3D, A3F, A3G, and A3H), a correlation between increased HPV DNA editing in the viral Ori-promoter region and expression of A3A, A3C, and A3H was reported in 2008 using a differential DNA denaturation polymerase chain reaction (3D-PCR) to selectively amplify AT-rich edited genome sequences at a lower denaturation temperature 82 °C [185]. Subsequent studies showed that A3A and A3B expression levels and their activities are highly increased in HPV-positive keratinocytes, cervical lesions, and cancer tissues [186,187,188], most likely due to the expression of viral E6 and E7. As p53 restricts A3B expression [189], viral E6-mediated degradation of p53 leads to the increased expression of A3B. HPV16 E7 stabilizes A3A protein by inhibition of cullin 2-dependent protein degradation [190]. Notably, the increased A3s induce HPV hypermutations throughout the viral genomes of HPV16, 52, and 58, and almost evenly on both sense and antisense strands of each analyzed viral genome [191], which appears predominantly in CIN1, less in CIN2/3 and even below the detection level in cervical cancer tissues [191]. However, other studies show A3-mediated C-to-T substitutions in the E7 [192] and E2 ORF regions [193,194,195], but more often in the URR region [196].

The increased expression of A3s in HR-HPV cancers has been considered as a source of oncogenic drivers to introduce point mutations in the host genome [186,197,198,199] and may contribute to intratumor heterogeneity [200]. On the other hand, A3s act as a restriction factor to reduce HPV infection [187,201,202] and promote viral clearance from the infection [203]. In contrast to A3s, A2, a transcriptional repressor [204], displays a decreased expression in CRPV-induced tumor tissues [205]. As A3s facilitate antiviral activities [206,207] and are highly elevated in HPV-associated cancers, the A3 levels in HPV-positive tissues might serve as a prognostic biomarker [195,202,208,209].

As stated above, A3s utilize ssRNA and sometimes ssDNA as a substrate for genome editing. The papillomavirus genome is a double-stranded DNA (dsDNA) and should not normally be editable by A3s [210,211]. It is speculated that the enriched A3 editing in the HPV16 URR could be a result of the ssDNA stage at the origin of replication and/or transcription from the URR region. A3A- and A3B-mediated mutations are mainly caused by the deamination of the lagging strand template during DNA replication [212,213]. The question of how an A3 editing complex competes in a very brief time with replicating or transcription machinery on a ssDNA for editing without affecting viral DNA replication or transcription remains to be understood. Further hard evidence or careful studies are needed to draw a conclusion about the likelihood that A3 is involved in HPV genome editing. Since the frequency of detected HPV editing is significantly lower than that detected for HIV-1 and HBV to be edited by A3s [196], and highly sensitive techniques, such as 3D-PCR or NGS, had to be used for detection, future studies are required to elucidate how A3s mechanistically affect HPV genome editing during viral DNA replication.

## 8. Remarks and Perspectives

It has been sixteen years since our past review [44] on papillomavirus genome structure, expression, and post-transcriptional regulation. We have made significant progress in dissecting HPV genome structure and function, finding more proteins and noncoding RNAs in the regulation of genome transcription and post-transcriptional process, and enabling keratinocytes’ long-term expansion in culture [214]. Since 2006, several HPV vaccines have been implemented by regulatory agencies, and, in 2018, at the 32nd International Papillomavirus Conference in Sydney, WHO called for action toward the elimination of HPV-induced cervical cancer by the year 2050 (https://www.who.int/reproductivehealth/call-to-action-elimination-cervical-cancer/en/, accessed on 1 March 2022). For HPV16, these advances include the characterization of a promoter P14 possible for E1 transcription [49] and a potential P1135 promoter for encoding E8^E2 [50]. Additional RNA cis-elements and RBPs required for regulation of RNA splicing and polyadenylation of HPV16 polycistronic RNAs were also identified [215]. The advances in HPV18 are the construction of a full HPV18 transcription map [38], subsequent observation of transcriptional repression and activation by host transcription factors [59,60,216], and the identification of new RNA cis-elements and protein splicing factors to begin our understanding of the regulated HPV18 RNA splicing [56]. The discovery of an ESS element in the E7 ORF region involved in the inhibition of HPV18 E6*I RNA splicing via binding host hnRNP A1 [56] opens up a new chapter to explore how and when the E6*I might be produced for translation of viral E7 protein during productive papillomavirus infection and in the course of high-risk HPV-induced oncogenesis. Similarly, this finding [56] led to the identification of a similar ESS element in the HPV16 E7 ORF region involved in the inhibition of HPV16 E6*I splicing and E7 protein expression [87]. Conceivably, the reduction in hnRNP A1 level, or blocking of hnRNP A1-ESS interaction in the HPV-infected cells, increases E6*I RNA splicing and E7 protein production, but decreases the expression of full-length E6 mRNAs and thus E6 protein production. To date, the expression level of E6*I RNA has been found to affect cervical Cancer Response to chemoradiation treatment (CRT) and might be viewed as a predictive biomarker of cervical cancer CRT [217].

The link between alternative RNA splicing in cancer development and progression has been one of the focal areas of research in the papillomavirus field since the 1990s. Alternative RNA splicing of viral polycistronic pre-mRNAs is essential for the expression of individual viral proteins from properly spliced RNA isoforms. Multiple RBPs, especially SR proteins and hnRNPs, have been identified as being involved in the regulation of HPV gene expression by binding to cis-elements in HPV pre-mRNAs. Among the host splicing factors facilitating the regulation of HPV16 and HPV18 RNA splicing, SRSF3 (SRp20) is perhaps the most remarkable host splicing factor, which not only regulates the early-to-late switch of HPV16 and HPV18 productive infection, but also prevents keratinocyte differentiation [56,79]. High SRSF3 expression in basal or undifferentiated keratinocytes prevents keratinocyte differentiation and promotes expression of viral E6 and E7, but not viral L1 (Figure 6b). Raft tissues and terminally differentiated cells in the papilloma and cervical tissues express very little or no SRSF3, thus supporting viral L1 expression and production of infectious virions [79,218]. Similarly, knockdown of SRSF3 expression in keratinocytes increases L1 production and keratinocyte differentiation [79]. Moreover, SRSF3 regulates the expression of 60 host genes and 182 splicing events in 164 genes [142,143] and functions as a proto-oncogene of which increased expression promotes cell proliferation, transformation, and tumorigenesis [138]. SRSF3 also regulates the expression of at least 20 cellular miRNAs and its expression is mutually co-dependent on SRSF1 [143], another proto-oncogenic host splicing factor [141] also involved in papillomavirus RNA splicing [91,93]. To date, many drugs targeting the splicing factors or oncogenic spliced RNA isoforms are under active development [219,220,221]. Seeking for splicing-targeted small molecules or chemicals specific to HPV-related cancers would be a future research focus.

While recognizing our past achievements, we remain puzzled by our lack of knowledge of how viral E1, E2, E5, and L2 are produced in the context of the HPV16 or HPV18 genomes, although their functions have been studied extensively by artificial overexpression of each protein from a mammalian expression vector with a foreign promoter. In the context of the HPV16 or HPV18 genomes during HPV infection, each of these genes is expressed along with other genes from a shared early or late viral promoter. Thus, the corresponding ORF of each gene is in the middle or last part of a polycistronic RNA. Considering that RNA 5′ cap-mediated ribosome scanning is from the RNA 5′ to 3′ direction and HPV RNA transcripts lack a ribosomal internal entry site, RNA splicing of the polycistronic RNAs removing or disrupting the upstream ORF(s) has been a plausible mechanism for the expression of E1, E2, E5, and L2 (Figure 3 and Figure 4). However, the E1 ORF and L2 ORF span over a respective intron (Figure 3 and Figure 4), of which splicing will disrupt their ORF integrity and abolish the production of the full-length protein. So far, we have no clue as to how these two introns could escape recognition by cellular splicing machinery, and, if they do, under what circumstances? In the context of the HPV16 and HPV18 genomes, L2 expression is more challenging, as it has to read-through the pA_E_ site on the intron 2 retained late transcripts. Whether viral E2 inhibits recognition of the pA_E_ site on the late transcripts by host polyadenylation machinery and promotes the read-through into the late region during HPV late infection [108] remains to be carefully explored.

Protein translation is the last step in gene expression, involving 5′ cap-mediated ribosome binding to a translation initiation codon AUG of the ORF. The canonical ribosome footprint is ~20–30 nt long from RNA [222,223,224,225]. A striking question in the HPV field, which remains to be addressed, is how such a short 5′ UTR upstream of the E6 ORF could provide enough space to fit ribosome occupancy and enable translation initiation. Because the majority of the HPV16 viral early transcripts are transcribed from the P97 promoter with a 5′ end at nt 97, and the HPV16 E6 translational start codon AUG is at nt 104, this leaves a very short 7-nt 5′ UTR for the HPV16 E6 to recruit a ribosome to initiate E6 translation. Similarly, the 5′ UTR of the HPV18 E6 ORF, of which the AUG starts at nt 105, is only 3 nt long if the viral early transcripts start from the promoter P102. This puzzle has been solved partially by recent identification of the HPV18 P55 promoter in transcription of the viral early transcripts in raft cultures and HeLa cells [38,58]. These findings indicate that the P55 promoter-derived early transcripts would be useful for translation of viral E6 and other proteins without any ribosome occupancy problem, whereas the P102 promoter-derived early transcripts are most likely responsible for translation of E7 or other downstream ORFs. Conceivably, seeking such a promoter upstream of the P97 promoter in HPV16 genome expression would be an important priority.

Non-coding RNAs (miRNAs and other small RNAs, lncRNAs, circular RNAs, etc.) have revolutionized our understanding of the complexity of the transcriptome involving every part of modern biology [226,227,228]. Although non-coding RNAs are not part of this focused review and the HPV genome does not encode any authentic viral miRNAs [229], and produces only a scarce amount of circE7 RNA (0.4 copies per CaSki cell) [48], high-risk HPV infections induce aberrant expression of host miRNAs and lncRNAs with oncogenic or tumor-suppressive functions that may contribute to HPV infection-induced carcinogenesis [22,182,229,230,231,232,233,234,235,236,237]. The altered expression changes in miRNAs and lncRNAs during HPV infections are induced mainly by viral oncoproteins E6 and E7 [22,229,231,236]. The altered expression of miRNAs and lncRNAs may affect the expression and function of RBPs, and thus HPV RNA processing, as each miRNA may have an average of ~300 conserved targets [226,238], and many lncRNAs function as RNA-binding protein sponges [239,240]. Moreover, the miRNAs and lncRNAs with altered expression may serve as possible biomarkers for diagnosis and prognosis of HR-HPV infection-induced lesion progression [229,234,236,241,242]. Looking forward, the study of non-coding RNAs and HPV interactions in HPV-infected cells holds great promise for further understanding the life cycle and pathogenesis of this important virus.

## Figures and Tables

**Figure 1 ijms-23-04943-f001:**
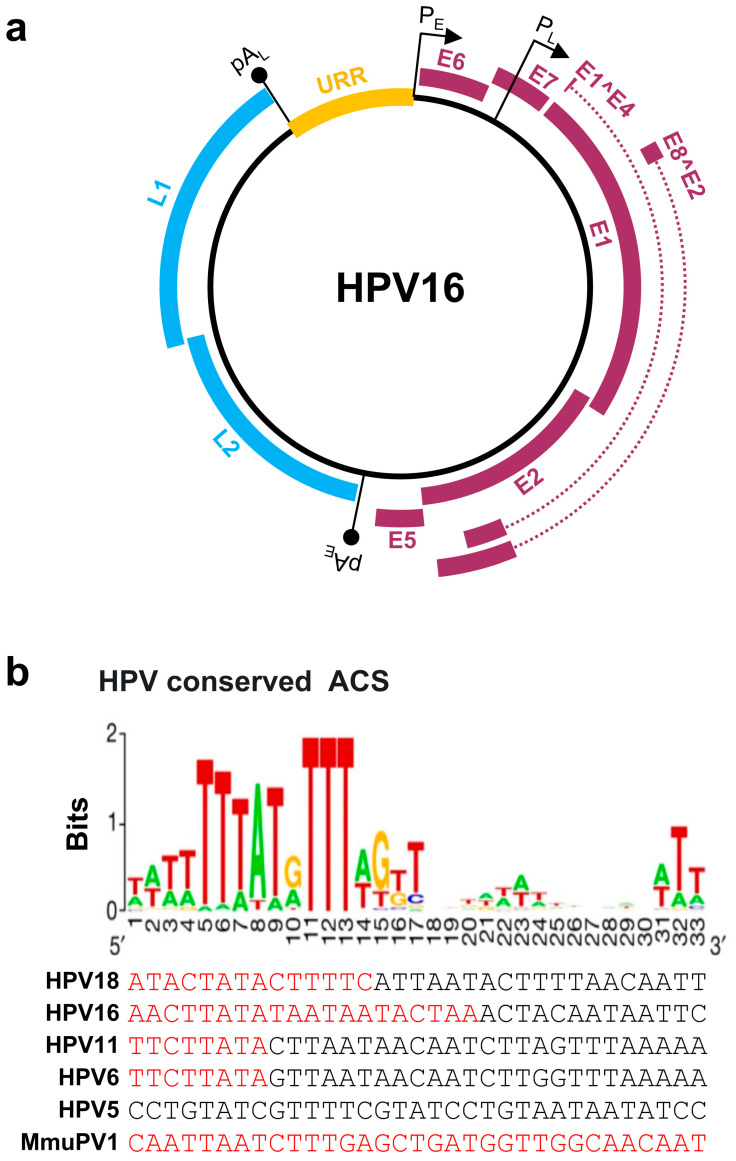
Papillomavirus genome structure, annotated ORFs, and conserved ACS in the origin of replication (Ori). (**a**) HPV16 genome and annotated ORFs. The non-structural proteins from the early region of the genome, including E1, E2, E1^E4, E5, E6, E7, and E8^E2, are shown in purple. The viral capsid proteins, L1 and L2, from the late region of the genome are shown in blue. URR (yellow), upstream regulatory region. P_E_ and P_L_ mark the early and late promoters, and pA_E_ and pA_L_ stand for the early and late polyadenylation sites. (**b**) Conservation of the mammalian ARS consensus sequence (ACS) [16] in the Ori of selected papillomavirus genomes. Shown in this panel are conserved bases with position weight matrix (sequence logo bits) of the 167 predicted ACS elements [16] in comparison with the viral Ori sequences from the selected viral URR tail (red)-head (black) regions. Adapted with permission from Ref. [16]. Copyright 2018 Springer Nature. ARS—autonomously replicating sequence.

**Figure 2 ijms-23-04943-f002:**
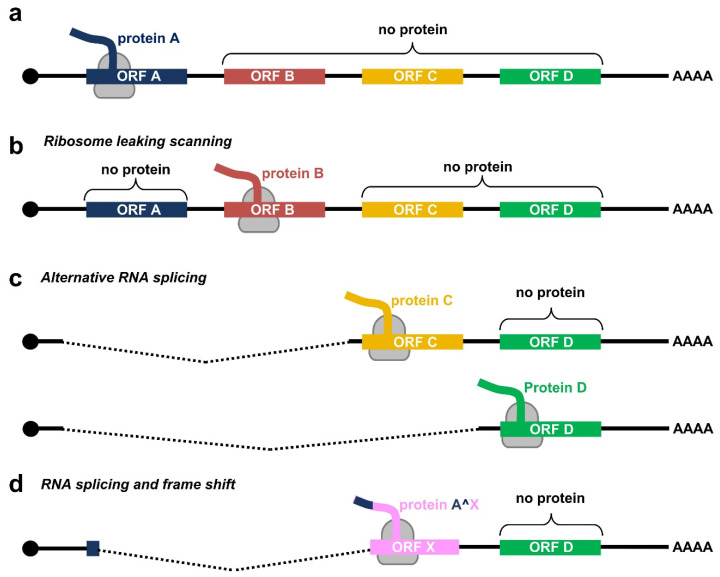
Translation modes of HPV proteins from polycistronic viral transcripts. (**a**) Diagram of a polycistronic mRNA harboring multiple ORFs (ORF A–D), of which only the first ORF A is translated by eukaryotic ribosomes (in grey) into a functional protein. (**b**–**d**) Strategies utilized by papillomaviruses to express viral proteins from polycistronic transcripts by possible ribosome leaking scanning (**b**), alternative splicing (**c**), or usage of alternative frames created by RNA splicing (**d**).

**Figure 3 ijms-23-04943-f003:**
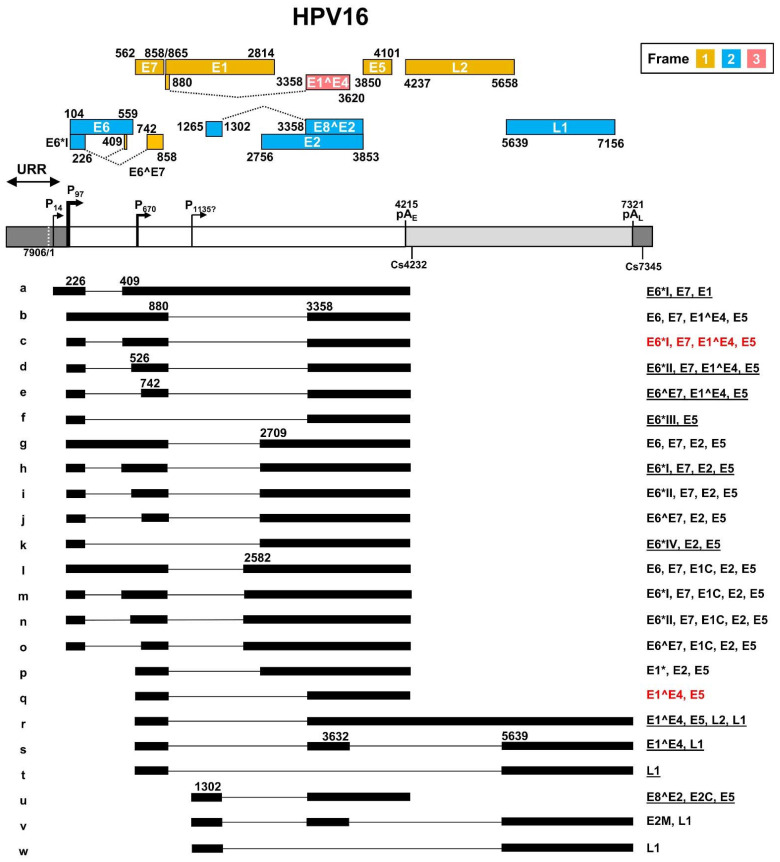
The transcription map of HPV16 updated from Ref. [44]. The HPV16 genome in a linear form diagramed as a bracket line consists of the upstream regulatory region (URR, dark grey box), early region (white box) and late region (light grey box) with nucleotide positions of viral promoters (P), early (pA_E_) and late (pA_L_) polyadenylation signals, and polyadenylation cleavage sites (Cs). The diagram is not to scale. The predicted viral ORFs with nucleotide positions in the viral genome and the corresponding frame used for their translation are shown above. Below the linearized genome are the mapped viral transcripts with identified exons (thick lines) and introns (thin lines). The numbers represent the nucleotide positions of splicing donor sites (5′ splice site) and splicing acceptor sites (3′ splice site). On the right are coding potentials for each transcript, with the most abundant (>60%) transcripts labeled in red color, and less abundant, but detectable ones (<5%), being underlined.

**Figure 4 ijms-23-04943-f004:**
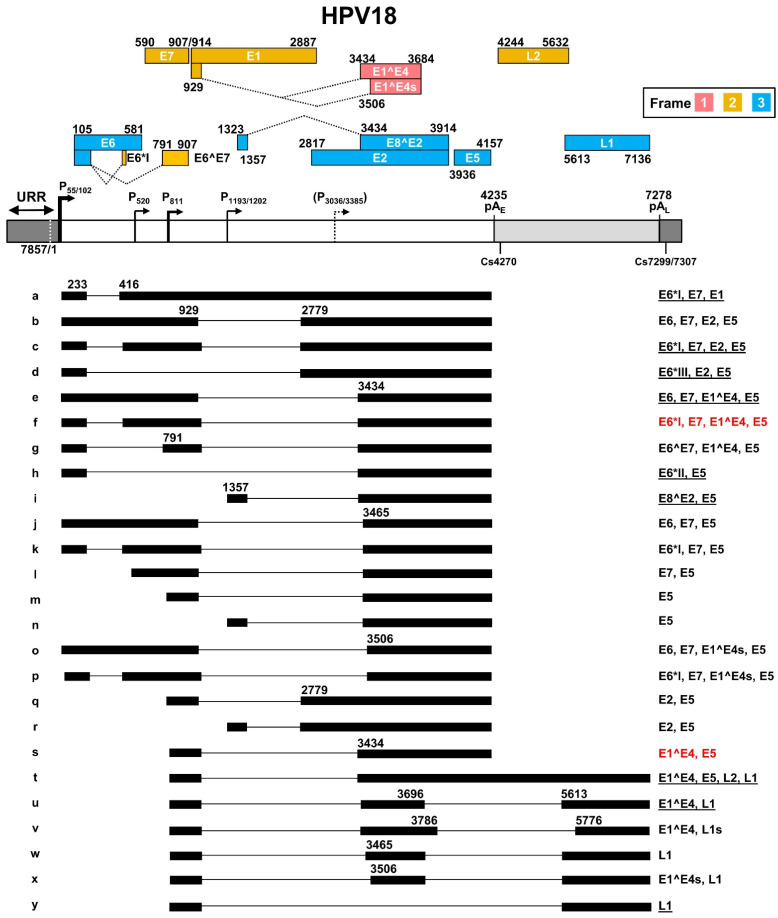
The transcription map of HPV18 updated from Ref. [56]. See the HPV16 transcription map for other description details.

**Figure 5 ijms-23-04943-f005:**
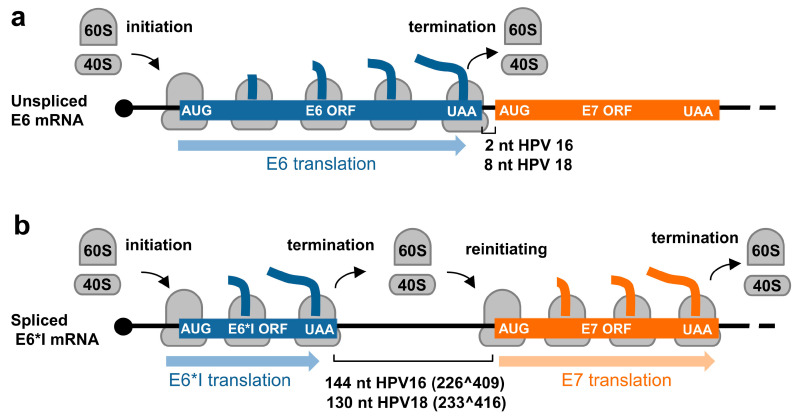
Regulation of E6 and E7 translation from HPV16 and HPV18 polycistronic E6E7 transcripts by alternative RNA splicing. (**a**) Diagrams showing E6 translation from the unspliced E6E7 RNA and (**b**) E7 translation from the spliced E6*I RNA by translation re-initiation.

**Figure 6 ijms-23-04943-f006:**
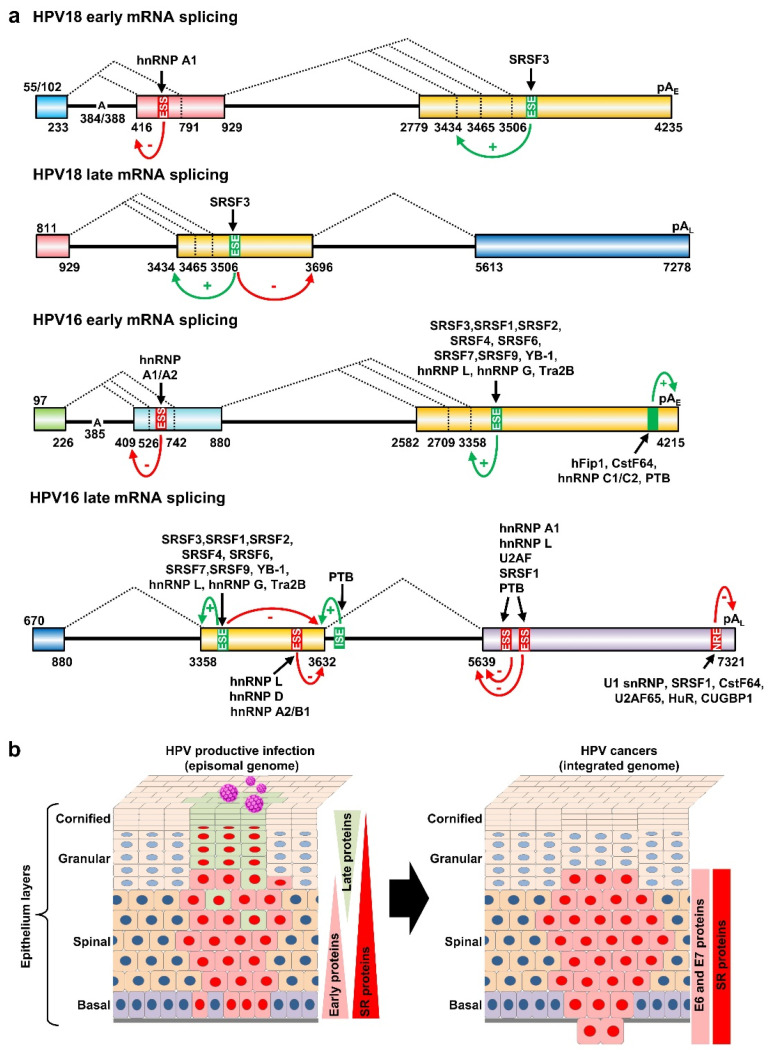
Regulation of HPV18 and HPV16 RNA splicing and polyadenylation by RNA cis-elements and host RBPs. (**a**) Diagrams (not to scale) showing the exonic (colored boxes) and intronic (thick lines) regions of HPV18 and HPV16 transcripts. The major alternative splicing events are depicted by dashed lines with the nucleotide positions of splicing sites shown below. Exonic splicing enhancers (ESE), exonic splicing silencers (ESS), intronic splicing enhancers (ISE), and negative regulatory elements (NRE) are shown in green for positive (+) and in red (−) for negative regulation. The black arrows show the host RBPs binding to individual cis-elements. The mapped branch site A in the intron branch point sequence is indicated by its genome position in the viral early transcript. (**b**) Depiction of HPV-infected epithelium with productive HPV infection (on the **left**) or with HPV-transformed cells invasive of the basement membrane (on the **right**). HPV-infected cells expressing viral early proteins are shown in red and the cells expressing HPV late proteins in light green. The changes in the expression of viral proteins and host splicing factors (SR proteins) are on the right of each diagram.

**Table 1 ijms-23-04943-t001:** The genomes and transcription regulatory elements of selected HPVs and animal papillomaviruses.

Species	Genome Size (bp)	Size of URR (bp)	No. of Major Promoters	No. of Splice Sites	No. of pA Sites	No. of CG Boxes	No. A3 Box TTC
HPV18	7857	825	2	11	2	171	73
HPV16	7906	853	2	11	2	112	73
HPV11	7931	756	3	8	2	154	56
HPV6	7996	806	3	7	2	160	77
HPV5	7746	478	4	10?	2	186	96
MmuPV1	7510	609	5	8	2	197	95
CRPV	7871	677	5	8	2	232	98
BPV-1	7946	940	7	11	2	181	89

MmuPV1—Mus musculus papillomavirus 1 or mouse papillomavirus type 1, CRPV—cotton tail rabbit papillomavirus, BPV-1—bovine papillomavirus type 1, URR—upstream regulatory region, pA—polyadenylation signal, and A3—APOBEC3. A3-binding sites were counted based on the studies [17,18]. All the reference genomes analyzed were downloaded from the Papillomavirus Episteme website (PaVE) (https://pave.niaid.nih.gov/, accessed on 1 March 2022).

## Data Availability

Not applicable.

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
