# Peer review of "HPV16 and HPV18 Genome Structure, Expression, and Post-Transcriptional Regulation"

_ijms, 2022, doi:10.3390/ijms23094943_

Round 1

Reviewer 1 Report

The review by Yu, Majerciak and Zheng “(ijms-1670646) gives an interesting and updated summary of HPV16 and HPV18 genome structure, expression, and post-transcriptional regulation. The authors also described DNA and APOBEC-mediated genome editing in HPV infection.

The manuscript is well written, detailed and includes demonstrative figures.

I would recommend publishing this review in IJMS. However, a few points should be considered:

  1. Since HPV16 and HPV18 are high-risk HPV types that can cause invasive cancers, a brief introduction on the mechanism of HPV-induced carcinogenesis should be included. It would be helpful for the readers.
  2. The quality of some figures should be improved, e.g., Figure 2 - ORFC is not visible, Figure3/4. - Frame 1(E6, E7 and L1) is not visible. In all cases, the background is too dark. Figure 6 - ESS, ESF are not visible. The font is too small.
  3. The name of chapter 5: Viral RNA polyadenylations and its regulations, should be changed as it suggests that authors describe this process in different viruses, not only in HPV
  4. A conclusive paragraph should be included

Author Response

Authors’ point-by-point responses to reviewers’ comments

Reviewer 1:

The review by Yu, Majerciak and Zheng “(ijms-1670646) gives an interesting and updated summary of HPV16 and HPV18 genome structure, expression, and post-transcriptional regulation. The authors also described DNA and APOBEC-mediated genome editing in HPV infection.

The manuscript is well written, detailed and includes demonstrative figures.

I would recommend publishing this review in IJMS. However, a few points should be considered:

  1. Since HPV16 and HPV18 are high-risk HPV types that can cause invasive cancers, a brief introduction on the mechanism of HPV-induced carcinogenesis should be included. It would be helpful for the readers. 

Yes. A brief introduction was made in this revision in the introduction line 38-41.

  1. The quality of some figures should be improved, e.g., Figure 2 - ORFC is not visible, Figure3/4. - Frame 1(E6, E7 and L1) is not visible. In all cases, the background is too dark. Figure 6 - ESS, ESF are not visible. The font is too small.

 We changed the color of the text from black to white for better visualization in figures 2-4, and enlarged the font (ESS/ESE) in Figure 6.

  1. The name of chapter 5: Viral RNA polyadenylations and its regulations,should be changed as it suggests that authors describe this process in different viruses, not only in HPV

 we changed the name of chapter 5 from “ Viral RNA polyadenylations and its regulations” to “Papillomavirus RNA polyadenylations and its regulations” in line 497.

  1. A conclusive paragraph should be included

The remarks and perspectives in this review cover the requested conclusion and also leave many open questions for further discussion and future investigation. We thought this section should be enough for what the reviewer is requesting. In addition, we provide a new paragraph in this section discussing the challenging question on how a short 5’ UTR of the HPV16 and HPV18 E6 ORF could fit ribosome binding and translation initiation for E6 translation (line 792-808). We hope the reviewer is ok with the addition.    

Reviewer 2 Report

RECOMMENDATION:
Accept after Minor Revision

SUMMARY:
In this submission, Yu et al. provide an excellent update on transcriptional regulation of human papillomavirus (HPV) types 16 and 18, which together are responsible for the majority of HPV-associated cancers. Particularly valuable is the documentation of numerous transcripts with varying coding potential in Figures 3 and 4, which I anticipate will become a widely cited resource. To my mind, the paper is excellent and there is no question it should be accepted. However, I’d like to see more effort given to organization, clarity and accuracy of language/English, and explanation of terminology. I’d also like to see more acknowledgement and explanation of precedent of conventions, including explicit explanations of the most significant differences between this paper and earlier reviews and PaVE (Papillomavirus Episteme; https://pave.niaid.nih.gov/). Finally, I’d like the authors to provide a comprehensive list of their annotations. This would make the results of this paper immediately usable (e.g., in bioinformatics analyses), saving readers the work of tediously compiling information from the text and figures, and would therefore dramatically increase the impact of this work.

MAJOR POINTS
1. Provide comprehensive annotations as a text or Excel-readable file. This should include transcripts, promoters, splice sites, enhancer/silencer elements, and any other relevant genomic features mentioned in the manuscript. This can be either as a supplementary table or a GTF file -- or, better, a dynamic, versioned online resource (GitHub-hosted file?). Along these lines, there seem to be important omissions in the paper, e.g., the authors state there is a 65-nt AC-rich splicing enhancer element promoting the usage of splice site 3358 (line 415) but do not give its coordinates. The lack of such a list really limits the impact of this paper in its current form -- the greatest value of this paper would lie precisely in the ability of readers to immediately use and apply the expertise of the authors, but this is very difficult without such a list.
2. Acknowledge, explain, and cite any precedent or conventions used. For example, what are the most significant differences between these updated transcript maps and those which can currently be found at PaVE? (Please give the LAST ACCESSED DATE for these resources when they are mentioned on line ~161 or 165.) Also, for HPV16, why do some PaVE maps and that of Schwartz 2013 (Appendix I) not include a transcript with the potential to encode E1? Indeed, perhaps Schwartz 2013 is an important previous review worth citing? https://pubmed.ncbi.nlm.nih.gov/23706315/
3. The authors state that there are ~200 HPV types (e.g., line 27). This is out of date; there are currently 448 HPV types documented at PaVE. Note that there has been a large increase since 2018 due to the discovery of numerous Beta- and Gamma-PV types in studies of immunodeficient individuals; for a review, see McBride 2022 https://pubmed.ncbi.nlm.nih.gov/34522050/
4. Figure 1b: if I’m understanding correctly, the black sequence shown above the alignment is WRONG, or has not been explained. For example, the sequence logo (above) implies consensus nucleotides of TATT… whereas the alignment implies consensus nucleotides of NTCT… — please either explain what the black sequence represents, omit it, or correct it. Also explain if there is a reason for the order of sequences in the alignment.

MINOR POINTS
1. I felt the usage of the term ‘intron’ could use additional background and clarification. Specifically, I think the term may be confusing to those (like me!) who have previously view exons and introns to be properties of genes, not of multi-gene transcripts. I think this was due to my ignorance, but I doubt I’m alone, and therefore feel it would benefit the paper to add a brief clarification. For example, it could be pointed out early in the manuscript that this term is being used to describe portions of RNA that are excised to produce the mature mRNA, even if those excised portions occur BETWEEN DIFFERENT genes, and even if the resultant transcripts encode MULTIPLE protein products. I realize this is somewhat covered in the Abstract, but a more explicit explanation would be valuable, e.g. at the beginning, or around line 152.
2. Could the authors clarify the function (IF ANY) of upstream start codons in E6? For example, for HPV16REF, there is an upstream in-frame start codon at position 83, whereas the annotated start is at 104. Is it possible these are ever used (e.g., in transcripts arising from p14), or is there any relevant evidence?
3. Line 36: change ‘15’ to ‘~15’ because the exact number is not agreed upon.
4. Line 55 and throughout: I strongly suggest following the widely-used PaVE convention of naming the upstream regulatory region as the URR, not LCR (long control region). Beyond consistency, LCR can be confusing because this acronym is also often used in genomics for ‘low-complexity region’, which in cases of misinterpretation gives the diametrically wrong picture of the function-rich URR. Moreover, the non-coding region (NCR on line 57) is often used for the space between E5 and L2, not the URR. Thus, I would OMIT the ‘NCR’ in the text -- it’s never used again.
5. Table 1: explicitly state what reference sequences these refer to (e.g., are they simply the ones at PaVE?). This allows reproducibility and also acknowledges the fact that within-type variants could differ.
6. Figure 1a: consider placing the labels of E8^E2 and E1^E4 above where those ORFs BEGIN at the top left; the current position of their labels is confusing.
7. Line 142: consider changing to “overlapping with E2 and E1, respectively” -- this clarifies that E1^E4 has an out-of-frame overlap with E2 (i.e., E4 is within E2) and that E8^E2 has an out-of-frame overlap with E1 (i.e., E8 is within E1).
8. Figure 3, top: please explain the reason that there are no E6* products shown for HPV16, as they are on PaVE and also in the HPV18 section including Figure 4. Please also explain if E6* products are thought to have any biological function or role.
9. Figure 3, top: the E1 portion of E1^E4 should align with the beginning of E1, but it currently appears to overlap E7.
10. Figure 3 and 4: on the right, would it be possible to indicate (bold or underline?) the MOST LIKELY protein(s) produced from each transcript (for example, the likeliest product is bold or underlined); as well as the likely ABUNDANCE of each transcript (for example, low/medium/high/unknown)?
11. Is there no TATA box in HPV16? This is described for HPV18.
12. Lines 190-2 state that the MAJORITY of late pre-mRNAs are spliced or double spliced, whereas lines 449-50 state that only a SMALL PROPORTION of late transcripts are double spliced. Please clarify.
13. Lines 209-10: it is unclear what “respectively” is referring to, i.e., an ordered list of two elements does not appear earlier in the sentence.
14. Figure 5a: consider indicating where the splice occurs.
15. Line 375: for “223^409”, should this be 226 instead of 223? Please clarify.
16. Lines 417-21 contain a long run-on sentence that is difficult to read. Please simplify or break up.
17. Figure 6: why are some splices not shown for HPV16, e.g., 226^3358? Explain how those shown were selected.
18. Lines 654-659 are impossible to understand.
19. Lines 754-757 are difficult to understand.
20. Line 779: is ‘transcriptome’ meant? (“RNA world” implies a whole range of hypotheses about the origin of life.)
21: Lines 792-4: the last sentence of the paper is full of English problems (not a strong ending!). For example, consider “Looking forward, the study of noncoding RNAs and HPV interactions in HPV-infected cells holds great promise for further understanding the life cycle and pathogenesis of this important virus.”

PROOFREADING/ENGLISH
Line 12: change ‘transcription and’ to ‘transcriptional and’
53: change to ‘DNA genome of length ~8kb’
70: change to ‘The HPV early’
72: change to ‘accessory’; delete unnecessary space
78: change to ‘gate cell cycle checkpoints’ (no ‘the’)
82: change to ‘the HPV genome’
91: change to ‘The number of identified promoters varies’
99: should it be ‘a late promoter’ (>1) or ‘the late promoter’ (just 1)?
108: ‘Shown’
112: I cannot understand “papillomaviruses in this genomic gene organization inherently leads…”
150: change ‘are covered by ORF’ to ‘encode an ORF’
151: delete ‘being translatable’
166: ‘an updated’
170: delete ‘diagramed’ (unnecessary and a typo)
190: “The majority”
197: change ‘would not make’ to “is not thought to have”
198: consider changing ‘presumable’ to ‘putative’
200: change “whereas” to “whereas the”
203: “The HPV16 genome”
216: “substantially overlap introns”? (“are spanning over” is awkward English and vague). Or do you mean that they overlap splice sites?
218: “remains a mystery”?
233: delete unnecessary space
256: add closing parentheses
258: add a reference to Figure 3 in this caption
270: delete ‘The’ in ‘The intron 1’
273: delete ‘the’ in ‘the intron 2’
287: delete unnecessary space, change to “is subject to regulation by”
303: “confined to”
352: is “Ample evidence” meant?
364: “where one heptamer”
378: “However, downstream protein target(s)”
407: the ‘E’ in ‘pAE’ should be a subscript
409: delete ‘the’ in “the intron 2”
425: delete ‘the’ in “the intron 2”
453: delete ‘the’ in “the intron 2”
454: “involves both”
464: consider starting a new paragraph here
468: delete unnecessary space
470: move the lone word “investigated” beneath figure, to next page
475: “splicing enhancers”
483: move header to next page
485: delete ‘the’ in “the RNA stability”
518: delete ‘the’ in “the pAE”
562: is ‘inversely’ meant?
638: “the DNMT1 promoter”
702: “a conclusion about the likelihood that A3 is involved in…”
723: “regulation of”
739: “identified as being”
740: delete ‘the’ in “the cis-elements”
751: “proto-oncogene”
758: delete “with cheers” (awkward and unscientific), change to “we remain puzzled”
774: “intron 2”
789: ‘sponges’?

Author Response

Authors’ point-by-point responses to reviewers’ comments

Reviewer 2:

SUMMARY:
In this submission, Yu et al. provide an excellent update on transcriptional regulation of human papillomavirus (HPV) types 16 and 18, which together are responsible for the majority of HPV-associated cancers. Particularly valuable is the documentation of numerous transcripts with varying coding potential in Figures 3 and 4, which I anticipate will become a widely cited resource. To my mind, the paper is excellent and there is no question it should be accepted. However, I’d like to see more effort given to organization, clarity and accuracy of language/English, and explanation of terminology. I’d also like to see more acknowledgement and explanation of precedent of conventions, including explicit explanations of the most significant differences between this paper and earlier reviews and PaVE (Papillomavirus Episteme; https://pave.niaid.nih.gov/). Finally, I’d like the authors to provide a comprehensive list of their annotations. This would make the results of this paper immediately usable (e.g., in bioinformatics analyses), saving readers the work of tediously compiling information from the text and figures, and would therefore dramatically increase the impact of this work.

MAJOR POINTS
1. Provide comprehensive annotations as a text or Excel-readable file. This should include transcripts, promoters, splice sites, enhancer/silencer elements, and any other relevant genomic features mentioned in the manuscript. This can be either as a supplementary table or a GTF file -- or, better, a dynamic, versioned online resource (GitHub-hosted file?). Along these lines, there seem to be important omissions in the paper, e.g., the authors state there is a 65-nt AC-rich splicing enhancer element promoting the usage of splice site 3358 (line 415) but do not give its coordinates. The lack of such a list really limits the impact of this paper in its current form -- the greatest value of this paper would lie precisely in the ability of readers to immediately use and apply the expertise of the authors, but this is very difficult without such a list.

Thank you for your suggestion! As many of these comments are reflected in the Fig. 3, Fig. 4, and Fig. 6,  we decided not to have these descriptions repeated again in a separate table in order to avoid redundancy. However, we did take the reviewer’s comments and summarized all the mapped cis elements and branch point sequences of HPV18 and HPV16 in a supplemental table S1. The AC-rich ESE mainly enhances 880^3358 splicing as described in this revision (line 425).

  1. Acknowledge, explain, and cite any precedent or conventions used. For example, what are the most significant differences between these updated transcript maps and those which can currently be found at PaVE? (Please give the LAST ACCESSED DATE for these resources when they are mentioned on line ~161 or 165.) Also, for HPV16, why do some PaVE maps and that of Schwartz 2013 (Appendix I) not include a transcript with the potential to encode E1? Indeed, perhaps Schwartz 2013 is an important previous review worth citing? https://pubmed.ncbi.nlm.nih.gov/23706315/

We have now described more in writing about the updates from the transcription map-1996 to transcription map-2006 and their difference from the one at PaVE (lines 168-175). Since the transcription map on newly described promoters and late transcripts from W12 cells at PaVE were not carefully verified from the publication, we decide not to include these in our updated transcription map. Instead, we added one sentence on this and another publications in lines 214-217 “for possible future inclusion” in this revision. We cited Schwartz’s updated review in 2020 (line 733-735), instead of Schwartz 2013, as this updated review covers more comprehensively of all his works than what he published in 2013 Virology review. For E1, we think the transcript a from P14 in our map encodes E1 after 226^409 splicing as it was carefully demonstrated in the cited article (Haugen, JVI, 2008).

  1. The authors state that there are ~200 HPV types (e.g., line 27). This is out of date; there are currently 448 HPV types documented at PaVE. Note that there has been a large increase since 2018 due to the discovery of numerous Beta- and Gamma-PV types in studies of immunodeficient individuals; for a review, see McBride 2022 https://pubmed.ncbi.nlm.nih.gov/34522050/

Thank you for the correction! We have updated the number in line 27 and cited McBride’s review article as well as two additional publications including Tirosh O., et al. Nat Med 24: 1815-1821, 2018; Pastrana DV., et al. mSphere 3: e00645, 2018.

  1. Figure 1b: if I’m understanding correctly, the black sequence shown above the alignment is WRONG, or has not been explained. For example, the sequence logo (above) implies consensus nucleotides of TATT… whereas the alignment implies consensus nucleotides of NTCT… — please either explain what the black sequence represents, omit it, or correct it. Also explain if there is a reason for the order of sequences in the alignment.

Sorry for the confusion. This revision has deleted the original ARS305 as a highly conserved ACS element immediately below the sequence logo. In addition, we revised our figure legend by stating “position weight matrix (sequence logo bits) of 167 predicted ACS” (line 113-116). The order of the papillomavirus sequences in the alignment is based on high consensus of the viral Ori to the ACS. HPV18 replication Ori is the best characterized Ori so far. 

MINOR POINTS
1. I felt the usage of the term ‘intron’ could use additional background and clarification. Specifically, I think the term may be confusing to those (like me!) who have previously view exons and introns to be properties of genes, not of multi-gene transcripts. I think this was due to my ignorance, but I doubt I’m alone, and therefore feel it would benefit the paper to add a brief clarification. For example, it could be pointed out early in the manuscript that this term is being used to describe portions of RNA that are excised to produce the mature mRNA, even if those excised portions occur BETWEEN DIFFERENT genes, and even if the resultant transcripts encode MULTIPLE protein products. I realize this is somewhat covered in the Abstract, but a more explicit explanation would be valuable, e.g. at the beginning, or around line 152.

Thanks! Done accordingly (lines 156-161).

  1. Could the authors clarify the function (IF ANY) of upstream start codons in E6? For example, for HPV16REF, there is an upstream in-frame start codon at position 83, whereas the annotated start is at 104. Is it possible these are ever used (e.g., in transcripts arising from p14), or is there any relevant evidence?

This AUG is unlikely useful because of it weak Kozak sequence context. Please see our revision in lines 230-235.

  1. Line 36: change ‘15’ to ‘~15’ because the exact number is not agreed upon.

Revised (line 36).

  1. Line 55 and throughout: I strongly suggest following the widely-used PaVE convention of naming the upstream regulatory region as the URR, not LCR (long control region). Beyond consistency, LCR can be confusing because this acronym is also often used in genomics for ‘low-complexity region’, which in cases of misinterpretation gives the diametrically wrong picture of the function-rich URR. Moreover, the non-coding region (NCR on line 57) is often used for the space between E5 and L2, not the URR. Thus, I would OMIT the ‘NCR’ in the text -- it’s never used again.

Thanks for the suggestion. We have revised our manuscript accordingly and changed the LCR to URR and omitted NCR in the text (lines 59 and throughout).

  1. Table 1: explicitly state what reference sequences these refer to (e.g., are they simply the ones at PaVE?). This allows reproducibility and also acknowledges the fact that within-type variants could differ.

We used the reference genomes from PAVE, and we have added one sentence under the table 1 (line 74-75).

  1. Figure 1a: consider placing the labels of E8^E2 and E1^E4 above where those ORFs BEGIN at the top left; the current position of their labels is confusing.

Thanks. Revised accordingly in the Fig. 1a.

  1. Line 142: consider changing to “overlapping with E2 and E1, respectively” -- this clarifies that E1^E4 has an out-of-frame overlap with E2 (i.e., E4 is within E2) and that E8^E2 has an out-of-frame overlap with E1 (i.e., E8 is within E1).

Thanks. Revised accordingly (line 145-148).

  1. Figure 3, top: please explain the reason that there are no E6* products shown for HPV16, as they are on PaVE and also in the HPV18 section including Figure 4. Please also explain if E6* products are thought to have any biological function or role.

Thanks. Fig. 3 has been revised accordingly. We added one sentence on “E6*I as a labile polypeptide and its function to be investigated” (lines 359-360).

  1. Figure 3, top: the E1 portion of E1^E4 should align with the beginning of E1, but it currently appears to overlap E7.

Thanks. Fig. 3 has been revised accordingly.

  1. Figure 3 and 4: on the right, would it be possible to indicate (bold or underline?) the MOST LIKELY protein(s) produced from each transcript (for example, the likeliest product is bold or underlined); as well as the likely ABUNDANCE of each transcript (for example, low/medium/high/unknown)?

Good comments. We have highlighted relative detectable transcripts, with most abundant transcripts marked in red color and less abundant (<5%), but easily detectable ones being underlined. 

  1. Is there no TATA box in HPV16? This is described for HPV18.

Done accordingly (line 188-191).

  1. Lines 190-2 state that the MAJORITY of late pre-mRNAs are spliced or double spliced, whereas lines 449-50 state that only a SMALL PROPORTION of late transcripts are double spliced. Please clarify.

Thanks. Clarified in this revision.

  1. Lines 209-10: it is unclear what “respectively” is referring to, i.e., an ordered list of two elements does not appear earlier in the sentence.

Clarified and corrected (line 224-225).

  1. Figure 5a: consider indicating where the splice occurs.

Done.

  1. Line 375: for “223^409”, should this be 226 instead of 223? Please clarify.

Corrected (line 386). Sorry for the typo.

  1. Lines 417-21 contain a long run-on sentence that is difficult to read. Please simplify or break up.

Done (line 425-432).

  1. Figure 6: why are some splices not shown for HPV16, e.g., 226^3358? Explain how those shown were selected.

Fig. 6 shows the major alternative RNA splicing. The 226^3358 splicing is not a major splicing event, but exon exclusion.

  1. Lines 654-659 are impossible to understand.

Revised (line 669-673).

  1. Lines 754-757 are difficult to understand.

Revised (line 769-772).

  1. Line 779: is ‘transcriptome’ meant? (“RNA world” implies a whole range of hypotheses about the origin of life.).

Revised accordingly (line 810). Thanks. 

21: Lines 792-4: the last sentence of the paper is full of English problems (not a strong ending!). For example, consider “Looking forward, the study of noncoding RNAs and HPV interactions in HPV-infected cells holds great promise for further understanding the life cycle and pathogenesis of this important virus.”

Great suggestion and revised accordingly (line 823-825).

PROOFREADING/ENGLISH
Line 12: change ‘transcription and’ to ‘transcriptional and’-revised. Line 12.
53: change to ‘DNA genome of length ~8kb’ -revised, line 57.
70: change to ‘The HPV early’-revised, line 76.
72: change to ‘accessory’; delete unnecessary space-revised, line 78.
78: change to ‘gate cell cycle checkpoints’ (no ‘the’) -revised, line 84.
82: change to ‘the HPV genome’  -revised, line 88.
91: change to ‘The number of identified promoters varies’ accessory -revised, line 96-97.
99: should it be ‘a late promoter’ (>1) or ‘the late promoter’ (just 1)? -revised, line 104-105.
108: ‘Shown’ -revised, line 113.
112: I cannot understand “papillomaviruses in this genomic gene organization inherently leads…” -revised, line 118.
150: change ‘are covered by ORF’ to ‘encode an ORF’ -revised, line 156.
151: delete ‘being translatable’ -deleted, line 158.
166: ‘an updated’ -revised, line 174-176.
170: delete ‘diagramed’ (unnecessary and a typo).
190: “The majority” -revised, line 202.
197: change ‘would not make’ to “is not thought to have” -revised, line 210.
198: consider changing ‘presumable’ to ‘putative’ -revised, line 211.
200: change “whereas” to “whereas the” -revised, line 213.
203: “The HPV16 genome” -revised, line 218.
216: “substantially overlap introns”? (“are spanning over” is awkward English and vague). Or do you mean that they overlap splice sites? -revised, line 236.
218: “remains a mystery”? -revised, line 238.
233: delete unnecessary space -revised, line 252.
256: add closing parentheses-revised, line 276.
258: add a reference to Figure 3 in this caption. -revised, line 278.
270: delete ‘The’ in ‘The intron 1’ -revised, line 291.
273: delete ‘the’ in ‘the intron 2’ -revised, line 294.
287: delete unnecessary space, change to “is subject to regulation by” -revised, line 309.
303: “confined to” -revised, line 325.
352: is “Ample evidence” meant? -revised, line 366.
364: “where one heptamer” -revised, line 375.
378: “However, downstream protein target(s)” -revised, line 389.
407: the ‘E’ in ‘pAE’ should be a subscript- revised, line 418.
409: delete ‘the’ in “the intron 2”-deleted, line 420.
425: delete ‘the’ in “the intron 2”-deleted, line 436.
453: delete ‘the’ in “the intron 2” -deleted, line 465.
454: “involves both” ” - revised, line 466.
464: consider starting a new paragraph here.
468: delete unnecessary space - revised, line 482.
470: move the lone word “investigated” beneath figure, to next page- revised, line 482.
475: “splicing enhancers” - revised, line 487.
483: move header to next page - revised, line 497.
485: delete ‘the’ in “the RNA stability” - revised, line 499.
518: delete ‘the’ in “the pAE” - revised, line 532.
562: is ‘inversely’ meant? - revised, line 576.
638: “the DNMT1 promoter” - revised, line 651.
702: “a conclusion about the likelihood that A3 is involved in…”- revised, line 716.
723: “regulation of”
739: “identified as being” - revised, line 754.
740: delete ‘the’ in “the cis-elements” - deleted, line 755.
751: “proto-oncogene” - revised, line 766.
758: delete “with cheers” (awkward and unscientific), change to “we remain puzzled” - revised, line 773.
774: “intron 2” - revised, line 788.
789: ‘sponges’? - revised, line 820.
